# The Cost of Reproducibility in Artificial Intelligence

## Abstract

**Background.** The reproducibility crisis has not left artificial intelligence untouched. Lack of documentation in published research can make independent replication an unnecessarily laborious task. We propose the *cost* of reproducibility as the labour required to reproduce a method and its results due to lacking documentation.

**Objectives.** We aim to quantify the cost of reproducibility to determine significant variation between venues. We hypothesise that studies published in venues with strict reproducibility requirements in the review process are less costly to reproduce.

**Methods.** We propose five dimensions of the cost of reproducibility and evaluate them on a scale of 1 to 10, using objective characteristics *e.g.*, availability of code, data, parameter values and experiment setup. We reviewed 918 papers published between 2022-2024 from AAAI, ICLR, ICML, IJCAI, JAIR, JMLR and NeurIPS.

**Results.** Machine learning conferences are up to $16.52\%$ less costly to reproduce than artificial intelligence conferences and $12.91\%$ than journals. Award-winning papers are not less costly to reproduce than average papers at the same venue.

**Conclusions.** By quantifying the reproducibility cost, we find that the effectiveness of reproducibility standards depends on community support and strict enforcement in the review process, to significantly lower cost. We encourage the publication of appendices and reproducibility checklists, and a low cost as a key criterion for paper awards to drive community changes with examples of best practices.

## 1 Introduction

For over a decade, across all fields of science, a crisis in reproducibility of research has been a persistent problem (Baker, 2016); ranging from social sciences (Schmidt, 2009) to physics (Junk & Lyons, 2020), chemistry (Ciriminna et al., 2024), biology (Tiwari et al., 2021), medicine (Moonesinghe et al., 2007) and other fields including artificial intelligence (AI) (Hill, 2017; Gundersen & Kjensmo, 2018; Hutson, 2018; Heil et al., 2021). Recently, Gundersen et al. (2025) attempted to reproduce the thirty most cited AI studies and found an average reproducibility rate of $50.0\%$, whereas Raff (2019) estimated the reproducibility rate for Machine Learning (ML[1]) studies to be $63.5\%$. This is shockingly low, yet not an unsolvable problem. In the social sciences, a crisis of reproducibility was assessed by Schmidt (2009), reporting that reproducibility is "of major importance and highly respected", yet rarely discussed. Korbmacher et al. (2023) determined that replication improved in the social sciences through positive structural, procedural and community-driven changes.

Reproducibility is paramount in science, being widely seen as a "cornerstone of the scientific method" (Moonesinghe et al., 2007; Simons, 2014); Popper (1934) states that "non-reproducible single occurrences are of no significance to science". The first recorded issue of reproducibility dates back to the 17th century and is described by Robert Boyle in Shapin & Schaffer (1985); an interaction between Boyle, Hobbes and Huygens concerning an experiment where Huygens observed an 'anomalous suspension' of water in a vacuum:

> Unless the phenomenon could be reproduced [..] then no one [..] would accept the claims [Huygens had] made [..]. The accomplishment of replication was dependent on contingent acts of judgment. One cannot write down a formula saying when replication was or was not achieved.

---

[1]We consider ML to be part of the broader field of AI (McCarthy et al., 2006).

At the time, there existed only two machines in the world necessary for the experiment, one in possession of Boyle and Hobbes, the other in that of Huygens, and yet the independent investigators were not able to reproduce Huygens's findings. In this case, Hobbes and Boyle invited Huygens to their lab to demonstrate (successfully) how to reproduce the phenomenon, a costly solution but necessary nonetheless. It was deemed that the "accomplishment of replication was dependent on contingent acts of judgement", *i.e.*, the documentation provided was insufficient for the independent investigators to reproduce the experiment. The reproducibility cost of the original experiment in this anecdote is an unrealistic solution. Requiring the presence of the original author is not only cost ineffective; to repeat this scenario today with Huygens's presence is impossible. Thus, it is obviously necessary to consider reproducibility whilst documenting scientific findings. This anecdote illustrates why reproducibility is so fundamental to the scientific method and the complexity of determining whether a result can be considered reproducible or not.

The reproducibility cost can be expressed in monetary terms through resource requirements. A famously expensive example from physics involves CERN's Large Hadron Collider (Brüning et al., 2012), a ten-year project with a 4.75 billion USD price tag (Knapp, 2012). Reproducing research that requires such an instrument would be unrealistic, unless it is made available to independent investigators, and may also imply other costs, in terms of labour intensiveness (acquiring access) or time (the resource is in high demand). The issue of resource requirements is becoming more prevalent in AI research, e.g., in the context of large deep learning models. Large language models (LLMs), such as Open AI's ChatGPT-4 and Microsoft's Copilot models, take several months to train on thousands of state-of-the-art GPUs. Tian et al. (2019) conducted a reproducibility study of AlphaZero (Silver et al., 2018) and found that "A single training run requires [..] days of training on thousands of TPUs, which is an unattainable level of compute for the majority of the research community". The authors further state: "When combined with the unavailability of code and models, the result is that the approach is very difficult, if not impossible, to reproduce, study, improve upon, and extend". High demand for compute can be accepted as an "irreducible cost" by the original authors, as it is in general not in their reach nor their responsibility to make these resources available, whereas lack of documentation is clearly a "reducible cost" that could be mitigated relatively easily.

In this work, we evaluate the reproducibility cost of 1061 AI studies from seven major publication venues across five dimensions. We seek to determine which documentation qualities and standards in the review process are effective for lowering reproducibility cost in empirical studies. We make recommendations to stimulate the community to write documentation that results in low reproducibility cost. By understanding what drives the cost, and subsequently reducing it, the reproducibility rates of 50.0% (Gundersen et al., 2025) and 63.5% (Raff, 2019) could be significantly increased.

## 2 BACKGROUND

Although scientists generally agree that reproducibility is fundamental to the scientific method (Moonesinghe et al., 2007; Simons, 2014), definitions found in the literature vary, and terms such as replicability and repeatability have been used as synonyms for reproducibility. Notably, over the past three decades, with the increasing prominence of computer science and automatisation, there has been a tendency to define reproducibility as re-computation (Claerbout & Karrenbach, 1992; Gent & Kotthoff, 2014; Schwab et al., 2000) – a definition we find too narrow, as it reduces the reproducibility of methods and results to a subset of the documentation and the execution of computer instructions. Buckheit & Donoho (1995) limited reproducibility to reproducing the figures of the original authors using their source code, which can be useful for comparison, but is only a single step in the analysis of the results, which itself represents only one part of the scientific method.

More recently, Raff et al. (2025) also drew attention to the problems of confusing terminology when it comes to reproducibility, specifically within ML research; 45% of the surveyed papers use the terms reproducibility, repeatability and replicability interchangeably. They proposed to expand the terminology with five categories, to support various topics that AI researchers wish to address in regards to reproducibility of research. They also elucidated the relationship between the terminology defined by the ACM and their proposed rigour types, to disentangle them from reproducibility. We share with this work the focus on separating reproducibility terminology.

We are not the first to address the "cost of reproducibility". Poldrack (2019) describes the costs as the efforts and struggles for early-career researchers (ECR) within the field of neuroscience to ensure the

reproducibility of their work. It highlights the the difficulties for ECRs to produce and write quality research that adopts more reproducible practices. These key actors within neuroscience, such as the need for resources or data as well as the impact of the "Publish or Perish" mentality, are present in our field as well and plays a detrimental role to the reproducibility of published research. The distinction between this work and Poldrack (2019) is that we consider 'cost' as the impact on independent investigators, in contrast to factors affecting authors to produce less reproducible research.

In this work, we use the term 'reproducibility' to refer to a general scientific concept, rather than limiting it to the field of AI or computer science, for which we find two possible standards: Goodman et al. (2016) and Gundersen (2021). Goodman et al. (2016) separates the definitions of reproducibility into three types, which partially overlap with the definitions in Gundersen (2021), who define three degrees of reproducibility. We find the definitions in Goodman et al. (2016) to be more ambiguous; it lacks any definitions of independent investigators, as well as the concept of what constitutes the documentation of the method. We also disagree with the reasoning that the search for 'truth' is the main motive for reproducibility. Based on Popper (1934), we argue it is the credibility of a method. Thus, based on Gundersen (2021) and Popper (1934), we formally define reproducibility in section 4 for our methodology. An extension of this section can be found in Appendix B.

## 3 REPRODUCIBILITY QUANTIFICATION

Gundersen & Kjensmo (2018) as well as Gundersen et al. (2018) conducted a quantification study regarding reproducibility. They used 16 Boolean and categorical variables over three categories, namely method, data and experiment, to analyse the state of reproducibility in two conferences over two years each. They reviewed in total four hundred papers, of which 325 were applicable for the study. The variables were engineered to the quality of documentation on the method, availability of data, results and code. Based on their degrees of reproducibility, the authors concluded that none of the papers are 'fully reproducible', but that there was evidence for a statistically significant improvement over time. Although similarities are present between our methodology and theirs, there is more expressivity in our study, as we consider the problem of the reproducibility cost using numerical scores rather than Boolean attributes. These scores allow for a more fine-grained assessment. Furthermore, we separate the experimental setup and implementation of the method entirely, whereas in Gundersen & Kjensmo (2018) the experiment category covers both.

Whether the features selected by Gundersen & Kjensmo (2018) directly translate into reproducibility is also not completely clear. Raff (2019) defined 26 features over three types (unambiguous, mildly subjective and subjective) and collected values for these from 255 papers, published between 2012 and 2018, that they attempted to reproduce. The authors collected various features, including the number of plots and tables in the paper, the availability of code, the specification of hyperparameters and author availability. Some of these features coincide with our method. They showed the correlation between the collected features and whether the paper was reproducible, and thus determined that ten features are significantly correlated. In contrast to our study, Raff (2019) focused on papers that have already been (attempted) to be reproduced and were selected on based on their own 'historical interest', thus making the population of papers reviewed not necessarily representative of published studies in ML. Secondly, we aim to quantify the *cost* of reproducibility, rather than marking papers as reproducible and irreproducible. Raff (2019) also explicitly states that they do not use source code as part of the documentation of a paper, whereas we argue it is a key element of the documentation.

In a more recent approach, Gundersen et al. (2025) presented a systematic replication of 30 AI studies (ten publications from 2012, 2014 and 2016 each). The authors selected the most highly cited publications and only included papers where either data or data and code is available, based on the reproducibility types of Gundersen et al. (2022). This yielded 22 papers to be studied, the results of which represents the reproducibility of the most impactful research. They found a shockingly low (partial) success rate of $50\%$ within a time limit of 40 working hours. The working hours measured in their study have an important link with our work, as it is a direct expression of labour, i.e. the effort required to reproduce. The average successful reproduction, including partial success, took the authors 34 hours out of a 40-hour time limit. They encountered problems during this process that increased the time cost, and 17 out of 20 encountered problems directly related to a lack of ambiguities in documentation. An extension of this section can be found in Section A.3 and Appendix C.

## 4 METHOD

Our use of the term 'reproducibility' is based on the following derivation from Gundersen (2021):

> *Independent* investigators are able to perform an experiment testing the same *hypothesis* and draw the same *conclusions*, using the *documentation* provided by the original investigators.

In this definition, the terms shown in italics should be understood as follows:

**Independent:** Investigators who have no conflict of interest with the original work and have no access to information limited to the original investigators.

**Hypothesis:** One or more of the hypotheses presented and tested in the original work. These hypotheses and the tests used to challenge them are commonly of a formal statistical nature.

**Conclusions:** Drawing the same conclusion can be based on producing the same outcome, performing the same analysis or arriving at the same interpretation of the analysis.

**Documentation:** Any work produced, written or otherwise, provided by the authors, including supplementary material of a publication. This can include, but is not limited to: appendices, code written or used by the authors, data and other (digital) information.

### 4.1 FIVE DIMENSIONS OF REPRODUCIBILITY COST

Although reproducibility is discussed widely within science, and many examples of quantification studies of reproducibility within AI exist (Gundersen & Kjensmo, 2018; Raff, 2019; Pineau et al., 2021; Berrar, 2024), the cost of reproduction has been addressed less. We define the cost by the lack of information in the documentation of methods and results that can significantly complicate reproducibility. The choice of this definition, as previously explained, is that we aim to identify factors that increase the required effort by independent investigators that can be mitigated by the authors. Other factors that can complicate reproducibility, such as requirements for computing resources, fall hence outside of the scope of our analysis. Thus, this is an indirect measure of cost; our rubric score represents the quality of documentation of a study to determine whether one study can be considered better documented than another. Using the above terminology, we propose implementation, data, configuration, experimental procedure and expertise as dimensions of cost.

**Implementation Cost.** *Given the documentation shared by the authors on a new method, how much effort would it be to re-implement the method from scratch?* The cost of implementing a method mainly considers publishing the source code used for the results of the study. By publishing the source code, or publishing the method as part of publicly available software packages, the authors can lower the cost of re-implementation, facilitating testing of the same hypotheses and enabling independent researchers to determine possible improvements in computational efficiency or alternative methods. Publishing the implementation can also be substituted, augmented or enhanced with other materials, such as pseudo code, practical descriptions or designs and diagrams. We investigate the cost from the perspective of re-implementation, to generalise better on implementation documentation.

**Data Cost.** *Given the data description in the documentation, how much effort would it take to either: find the same data set the authors used, or a similar data set and defend the comparability, **or** acquire one from scratch?* The cost of acquiring data when reproducing a method and results is one of the most persistent problems in the reproducibility of AI research. We acknowledge here that publishing data is not always possible, e.g. in fields such as medicine, yet aim to stimulate authors to present their work with publicly available data, to be able to test a comparable hypothesis. Furthermore, the method of data acquisition should be clearly documented such that independent investigators can explain the comparability to other data sets, or acquire one with similar characteristics.

**Configuration Cost.** *Given the (hyper)parameters, including semantic parameters, of the method: how much effort would it take to acquire the algorithm configurations used for obtaining the reported results, and compare them against their computation budget?* The acquisition of algorithm configurations and hyperparameter values is mainly a computationally costly endeavour, and the budgetary constraints under which they were achieved can yield substantial variation in the results. By documenting the used values and under which conditions they were acquired, independent investigators

can reproduce results efficiently, and defend why these results can be used to test the same hypothesis. This includes documenting semantic parameters, e.g. parameters related to the task not impacting outcome quality such as the number of output nodes in a neural network.

**Experimental Cost.** *Given the setup of experiments reported in the work, how difficult is it to set up a new experiment with the same procedure, similar to those presented in the original work?* The cost of setting up the same, or comparable, experiment is crucial for evaluating the method. This requires documentation on the strategy and workflow, which metrics are used, and how the results are used to test the hypothesis. Lack thereof may significantly complicate the outcome, lead to (statistically) incorrect results and conflicting conclusions.

**Expertise Cost.** *How much effort would it take to acquire the expertise required to reproduce the work independently relying solely on the available documentation?* Acquiring expertise to reproduce a study may vary both on the side of the independent investigators' experience and the complexity of the study. We unify this by considering the cost of the acquisition of the prerequisite expertise, independent of whether this has been done. This may be caused by the need for knowledge from specialised sub-fields, requirements of inter-domain knowledge in applications and the understanding of mathematical theory. This dimension is difficult to quantify using objective characteristics.

### 4.2 Assessing the cost

To quantify each dimension of reproducibility cost, we answer the questions with a score ranging from one to ten, where one indicates very low cost and ten indicates very high cost. For each paper, we limited the documentation to all materials supplied by the publisher (e.g. the paper itself and possible supplementary materials) and any external links directly provided there (e.g. a link to a GitHub repository). No other materials were searched for online, to ensure a fair and comparable evaluation for each publication. We decided on a positive approach for the study and assumed all required documentation was provided, thus each review starts with a cost of **one** for all dimensions.

We evaluate each dimension based on our guidelines that cover a wide range of topics allowing for the expression of a variety of topics across studies and subfields. Implementation cost is focused on practical method documentation, such as pseudo-code, diagrams, design choices and libraries or systems used for creating the implementation as well as verifying code presence/quality when available. Data cost is focussed on all datasets used and is averaged with a weight of one per dataset by default, but allowed reviewers to adapt these weights on a case-by-case basis. Datasets were evaluated on their descriptions regarding their task and statistics, with ample documentation requirements regarding acquisition when private. Synthetic data or simulated environments were evaluated separately, with a focus on their implementation and parameters. Configuration cost is focused on clear and structured explanations of what the authors considered parameters of their method, using for example pseudo-code or tables, what values were used in each experiment and how they were acquired. The importance of parameters is taken into account when estimating the cost. Experimental cost is focused on the metrics used and what the population represents, including the type of error bars and aggregation functions, and what evaluation strategy is used including possible parameters. Expertise cost is mainly focused on how many subfields are used to what extent, and how well these are introduced. A detailed summary of the guidelines can be found in Table 1, with each rule categorised per dimension and subset.

### 4.3 Paper sampling

We sampled studies from five major conferences and two major journals from AI and ML over the past three years (2022-2024); two general AI conferences, (AAAI and IJCAI), three ML conferences, (ICLR, ICML and NeurIPS), and one general AI and ML journal each (JAIR and JMLR). We selected these venues based on their broad scope, excellent reputation and high impact on the scientific community. Per venue, we reviewed about fifty papers per year, selecting award-winning papers and sampling the remaining papers uniformly at random. We reviewed each paper according to our guidelines, noting the implementation link and the amount of (public) datasets in each work as metrics. Only regular papers were included in the sampling, e.g. excluding survey papers and position papers, as they generally do not provide an empirical evaluation. We also excluded papers that were originally present in another venue to avoid over-representation (i.e., journal versions of conference papers). Theoretical papers were included, but purely theoretical papers (e.g. no empirical

| Dimension | Subset | Description | Cost |
|---|---|---|---|
| Implementation | Code | Bad readme (dependencies unclear, bad examples) | +1 |
| | Code | Lack of comments | +1 |
| | Code | Lack of repository structure | +1 |
| | No Code | No (subset) of code given | +4 |
| | No Code | No statements on framework/libraries/languages etc. | +4 |
| | No Code | Some details on framework/libraries/languages etc. | +[2,3] |
| | No Code | Extensive practical details including design choices | +1 |
| | No Code | Statements of implementations used for their method | +0 |
| | Overview | Pseudo Code, designs or architecture missing | +[1,2] |
| | Misc. | No implementation, but other key details provided | -[1,2] |
| Data | Meta | Link to dataset not provided | +1 |
| | Meta | Dataset citation not given and no link | +1 |
| | Details | Dataset description lacking | +1 |
| | Details | Dataset statistics lacking | +1 |
| | Acquisition | Dataset (partially) private | +[1,2] |
| | Acquisition | Private dataset with unclear collection strategy | +[1,3] |
| | Misc. | Data set is only named | +[0,5] |
| | Synthetic | Code not provided | +[0,5] |
| | Synthetic | Process/Task not described | +[1,2] |
| | Synthetic | Parameters not clear | +[1,3] |
| Configuration | Parameters | No clear overview/summary | +[1,3] |
| | Parameters | Values not specified | +4 |
| | Parameters | Values not clear per experiment | +1/2 |
| | Acquisition | Strategy not clear | +1 |
| | Acquisition | Budget not clear | +1 |
| Exp. Procedure | Strategy | Strategy not clear | +[1,2] |
| | Strategy | Strategy parameters not clear | +1 |
| | Metrics | Used metrics not clear | +[1,2] |
| | Metrics | Aggregation/population not clear | +1 |
| | Metrics | Population variation not clear | +1 |
| | Data | Split for training/testing unclear | +2 |
| | Results | Data (subset) unclear | +1 |

Table 1: A summary of the data collection guidelines. In brackets, we express a range of options. Each guideline is specified per dimension, and subset. Note that not each guideline is applicable for each study. When multiple datasets are used, the value is a weighted average, with weights set to one by default. The complete guidelines can be found in the supplementary material.

evaluation) were marked and excluded from our analysis as the reproducibility of purely theoretical work poses challenges of a different nature. More details on the method can be found in Appendix E.

## 5 ANALYSIS

In this section, we analyse the data from the perspective of venues and dimensions, and the reliability of our results. The dataset is summarised in Table 2, which shows that the amount of effective reviews (918) is lower than the total (1061) due to theoretical papers. We sampled award-winning papers specifically and analyse them separately in Section 5.3; in all other analyses, these papers were not considered. For all statistical tests, we used a standard significance level of $0.05$.

### 5.1 SECOND REVIEW

A limitation of our dataset is the lack of reviewing resources. Due to the difficulty of the task, the effort required for reviewing and the challenge of finding proficient reviewers, initially, only a single review was acquired for each paper by our first author. To mitigate this limitation, we acquired a second review for 46 (i.e. $5.01\%$) out of the 918 papers from independent researchers in our network of PhD candidates and post-doctoral researchers. We let our 14 volunteers select up to five papers for

|       | AAAI     | IJCAI    | ICLR     | ICML     | NeurIPS  | JAIR     | JMLR     |
|-------|----------|----------|----------|----------|----------|----------|----------|
| 2022  | 46 (5)   | 40 (10)  | 46 (4)   | 53 (5)   | 44 (6)   | 39 (11)  | 39 (11)  |
| 2023  | 48 (2)   | 43 (7)   | 47 (3)   | 51 (5)   | 48 (2)   | 28 (22)  | 39 (11)  |
| 2024  | 48 (2)   | 47 (2)   | 48 (2)   | 47 (2)   | 48 (2)   | 33 (17)  | 36 (12)  |
| Total | 142 (9)  | 130 (19) | 141 (9)  | 151 (12) | 140 (10) | 100 (50) | 114 (34) |

Table 2: A summary of the collected data, where the first number represents the number of applicable reviews and the number of excluded (theoretical) papers is represented in brackets. We collected 918 applicable reviews out of 1061 papers. JAIR has the highest theoretical rate (33.33%) and AAAI the lowest (5.96%). The number of reviews (including theoretical) for IJCAI is less than 150 due to a rejected review (conflict of interest), and slightly more for AAAI/ICML due to a correction of a sampling error of award-winning papers. JMLR had less than 50 publications in 2024 at the time of sampling and is thus lower than 150 in total.

|                        | MAD    | ICC    | 95% CI       | P-Value | Label     |
|------------------------|--------|--------|--------------|---------|-----------|
| Implementation         | 0.7609 | 0.9727 | [0.95, 0.98] | 0.0000  | Excellent |
| Data                   | 0.7826 | 0.9111 | [0.84, 0.95] | 0.0000  | Excellent |
| Configuration          | 0.9130 | 0.9348 | [0.88, 0.96] | 0.0000  | Excellent |
| Experimental Procedure | 0.7826 | 0.8072 | [0.65, 0.89] | 0.0000  | Good      |
| Expertise              | 2.0652 | 0.5445 | [0.18, 0.75] | 0.0048  | Moderate  |

Table 3: The intraclass correlation coefficient (ICC) over the first and second review with 95% confidence interval, P-Value and their interpreted labels based on Koo & Li (2016).

secondary review based on their expertise and personal interest, but were presented with a subset to select from to ensure spread across years and venues. We calculated the inter-rater agreement using the Intra-class Correlation Coefficient (ICC) (Kotz et al., 2005). Based on the guidelines of Koo & Li (2016), we applied model ICC3$k$ which calculates the reliability of our data labelling guidelines over a fixed set of $k$ raters. The labels of agreement from Koo & Li (2016) range over 'poor', 'moderate', 'good' and 'excellent'. Table 3 shows that the first three dimensions strongly correlate among reviewers, with an 'excellent' label. Experimental Procedure has a 'good' label, showing some disagreement but in general still a reliable result. Expertise receives a 'moderate' label, which shows the complexity of assessing this dimension and the disagreement among raters, indicating the need for tangible guidelines; expertise only had loose descriptions without objective features. The mean absolute difference shows that the reviewers vary in their assessment of expertise cost by 2.0652, a factor of 2.55 above the other dimensions. Even when accounting for human bias, this is a noisy dimension compared to the others. Thus, we omit expertise in our subsequent analyses to ensure representative and reliable results. Due to space limitations, we analyse the relation between our methodology and reproduction attempts from Gundersen et al. (2025) in Section A.3.

## 5.2 ANALYSIS PER VENUE AND DIMENSION

In Figure 1, our data is visualised per dimension and venue. For the first two dimensions, the percentage of studies containing public implementation or public datasets (the public rate) are indicated by dashed lines. We plot percentages of studies, with each column adding up to 100%, as the absolute numbers vary slightly (see Table 2). The median and quantiles are reported in Table 5. We consider costs in the range of 1–3 to be 'low', and 7–10 to be 'high'.

Figure 1 shows that the implementation dimension dominates the other dimensions; even NeurIPS, with a public rate of 80.17%, has 23.97% of studies with a cost of six or higher. Most venues have little middle ground and represent somewhat of a bimodal distribution, which is strongly demonstrated by JMLR for example; studies either score rather low or high and this split overlaps with the public implementation rate. This implies that studies that do not publish their implementation in general do not seek out to use other facets of documentation, to supplement the absence of code. The configuration dominates the data and experimental procedure dimension, but has a substantially lower distribution than the implementation dimension. For all venues, the majority of studies, third quartile (Q3), has a cost of five or lower. The data dimension has a lower distribution, with most venues having cost of up to four for Q3; this is however not as strongly correlated with the very high overall

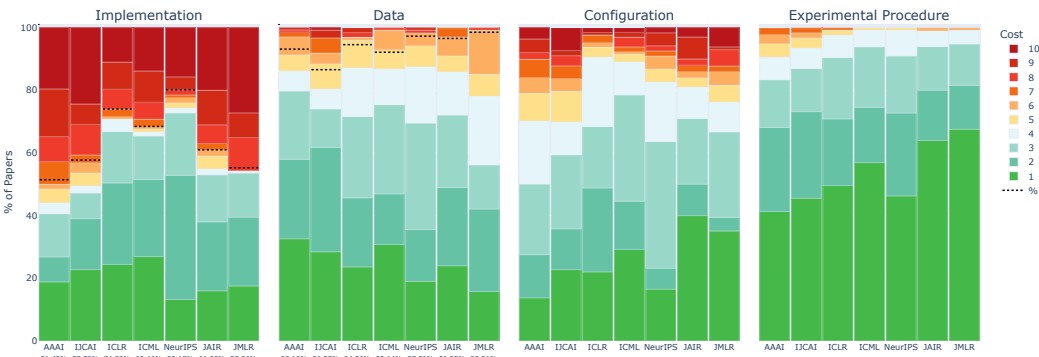

Figure 1: A summary of the collected dataset over seven venues. The first two dimensions have their respective percentages of public code or data plotted with a dashed line, the values are denoted in the caption of each bar. Over all empirical studies, we estimate that 70.52% of publish their implementation and 94.01% of used datasets are public. A tabular summary can be found in Table 5.

estimated public data rate of 94.01% as may be expected. We attribute this partially to an inflation of data cost, where authors tend to deem it unnecessary to extensively document commonly used datasets. This is discussed further in Appendix A. The experimental procedure is the best documented dimension. All venues have a cost of three or less in Q3. This is relatively unsurprising; experiment documentation is an integral part of the scientific method, within the field of AI and beyond.

In order to determine statistically significant differences between venues per dimension, we applied a normality test per venue, per dimension, the results of which can be found in Table 6. We found that none of our data are normally distributed. We thus opt for a one-sided permutation test to determine which studies cost statistically significantly less than another. Statistical tests are discussed futher in Appendix A. The results of the permutation tests can be found in Table 7.

**Implementation.** ICLR, ICML and NeurIPS have a significantly lower implementation cost than the other venues, showing a cost reduction between $-1.13$ and $-1.92$ compared to the general AI conferences and $-0.87$ to $-1.43$ to the journals. ML conferences tend to have better documented implementations than AI conferences, and papers published there are on average 16.52% less costly to reproduce. Compared to journals, we detected a cost reduction of 12.91% on average. Furthermore, we found that a substantial amount of studies follow a dark pattern: an estimated 6.72% of all implementation URLs are either empty or yield an error. This is not evenly distributed among venues: for general AI venues, the proportions range from 8.20% (JAIR) to 12.68% (AAAI), whereas ML venues range from 3.18% (JMLR) to 5.16% (NeurIPS).

**Data.** The data dimension is more costly for JMLR than the other venues, except for NeurIPS, although the measured differences are relatively small. As most venues in general do have rather low cost for the majority (Q3) of the studies, the practical significance of the measured differences seems to be low. JMLR has the highest public data rate of all venues (98.51%), yet surprisingly is the most costly in this dimension compared to the other venues. We attribute this partially to 'popular' (benchmark) datasets. We also find that the majority of studies who rely completely on private data (3.87% of all studies) have a high data reproducibility cost: 84.85% of private data studies have a cost of six or higher, mainly due to the collection strategy not being clearly documented. This could be mitigated by authors creating a datasheet, for example, as suggested by Gebru et al. (2021).

**Configuration.** ICLR and ICML have the lowest configuration cost compared to other venues, being significantly less costly than all other venues except for JAIR. JAIR has significantly lower cost than both general AI conferences and NeurIPS. Lastly, JMLR is less costly than AAAI. The biggest, and thus practically most impactful differences, are between ICLR/ICML and the other conferences: they are on average 11.70% less costly to reproduce. The most interesting outlier here is NeurIPS, a flagship conference of the ML community, yet more costly than ICLR/ICML by 0.71 and 0.66, respectively. As shown in Figure 2, on average, there is a weak correlation between public code and lower configuration cost of $-0.21$ overall. However, NeurIPS has a substantially weaker correlation ($-0.05$), indicating that the code is used less often to document the configuration than at other venues.

**Experimental Procedure.** AAAI has a significantly higher cost than all other venues except IJCAI, whereas IJCAI has a significantly higher cost than ICML, JAIR and JMLR. NeurIPS is significantly more costly than both journals. All distances are relatively small, ranging from $-0.27$ to $-0.66$. Thus, the practical impact of the differences are low. This is also shown by the fact that each venue has a low experimental procedure cost for the majority of studies (Q3) and that no studies with a high experimental procedure cost have been found. The studies we surveyed document this dimension well, and although there is room for improvement, this is marginal compared to the other dimensions.

### 5.3    Award-winning papers

We also reviewed 65 award-winning papers, i.e. best or outstanding papers at AAAI, IJCAI, ICLR, ICML and NeurIPS. JMLR and JAIR were excluded from this, as JMLR does not present such awards, and JAIR only has a single award winning paper in our time-frame. To test if award-winning papers are drawn from statistically significantly different distributions when compared to 'regular' papers, we applied the Kolmogorov–Smirnov test to check if they are drawn from the same distribution, and the permutation test to check if the cost is significantly lower. The results are found in Table 8. Our results suggest that only IJCAI shows a single significant difference according to the KS-Test, and only ICML best papers show a significant reduction in cost for the data dimension compared to average papers, an already well-documented dimension. We find no strong evidence that the cost of reproducing an award-winning paper is lower than an average paper published at the same venue.

## 6    Discussion & Limitations

According to our results, the two most ill-documented dimensions are implementation and configuration. Gundersen et al. (2025) found that the biggest bottleneck for reproducible research in AI is the accessibility of data and code. As the data dimension is generally well documented, with a high proportion of public datasets across the board, we conclude that the community has good standards here, although documentation for reacquiring private datasets is often lacking. The second most important dimension, according to their findings, implementation, is the least well documented of all. For example, 6.72% of all provided implementation links are either empty or broken. We find room for improvement for all venues, but papers published at ML conferences are significantly less costly to reproduce than those at other venues with respect to the implementation dimension.

The configuration dimension is the second least well dimension documented, and we find that publications at ICLR and ICML are less costly than those from other venues, except for JAIR. The most noteworthy outlier here is NeurIPS, a flagship of the ML community yet separated ICLR and ICML. Since all venues employ reproducibility standards in their paper submission process[2], the effectiveness of the approaches is interesting: NeurIPS has the strongest requirements, and since 2024 it is mandatory for all authors to append the reproducibility checklist to their main papers published there. This is in strong contrast to ICLR, which only "strongly encourages" a paragraph-long reproduciblity statement. In 2024, we found that 98.84% of accepted papers at NeurIPS included the checklist in the main paper, whereas only 15.79% of accepted papers at ICLR included the optional reproducibility statement. However, ICLR achieves similar results as NeurIPS in terms of documentation quality with far softer requirements for authors. This difference is hard to explain based on our dataset. An important factor that could contribute to this effect is that the participants of ICLR and NeurIPS are all part of the larger ML community. Thus, standards set out by either conference chairs or its participants can flow into other sub-communities as well. This effect can directly occur by means resubmission: papers rejected from one conference often find a home at another later, thus multiple sets of standards can affect the documentation quality of the paper. Another important factor for ICLR is its usage of a public review process, allowing both reviewers and the general public to read and respond to the submitted work. How often public interactions play a role in adapting the documentation of submitted work is unknown to us, or how authors adapt their work beforehand considering that it will be public *before* acceptance. This public process only applies to ICLR and warrants an analysis of its own to determine its role in reproducibility and whether it should be recommended to other venues.

---

[2]AAAI, IJCAI, ICLR, NeurIPS, ICML, JAIR and JMLR have checklists/standards for reproducibility.

The general AI conferences have the highest reproducibility cost across the board, with a few exceptions in the data dimension. These conferences have different reproducibility standards, and most importantly are the only venues that do not allow for appendices in accepted publications. Appendices are frequently used by authors to document key information; we found that, for the ML conferences in approximately 69.23% of the empirical studies the appendix contained information for one or more dimensions. However, the availability of appendices is not guaranteed to lead to better reproduciblity documentation; publications at JAIR and JMLR are less costly than those at AAAI and IJCAI in terms of configuration and experimental procedure, yet the differences are marginal. We argue that the longer format enables authors to provide in-depth documentation, but that the sheer availability of them does not serve as a stimulus for better documentation. Another reason that general AI conferences are more costly than others is that they might suffer, on average, from different standards in different sub-communities in AI, which should be studied in the future.

We also attempted to evaluate the expertise required for reproducing a paper but failed to do this reliably, as described in Section 5.1. This dimension was intended to represent a more intangible cost but turned out to be impractical, due to the lack of objective guidelines enabling objective assessment. Another limitation of our study is the sample size: as the process of evaluating the quality of documentation without relying on objective but simplified metrics, as for example done in Gundersen & Kjensmo (2018), is difficult to automate. Thus, we had to rely on human annotation, for a complicated and time expensive task resulting in only a small proportion of all published papers of each venue being analysed but argue that the sample size of 150 per venue is statistically robust. Note that our sample size is not proportionally adapted to each venue; this has no impact when analysing the venues individually, but have applied weighting when determining population wide trends. This is process is described more extensively in Section A.1.

# 7 CONCLUSION

We reviewed 1061 papers from seven major AI/ML venues over the past three years and analysed 918 on five proposed dimensions of reproducibility cost. We found significantly lower costs for ML conferences compared to other venues. We attribute this partially to stricter reproducibility standards, longer formats of papers through appendices, and, the interaction within the ML community at large. We recommend that venues require publicly visible reproducibility checklists for their submissions and allow appendices in publications to enable in-depth documentation. We also find that these measures independently are not effective, as contrasted by ICLR and NeurIPS as well as the general AI venues and journals, and that community changes are needed as well. While, in principle, this could be achieved through a stricter review process, this may be unrealistic in practice, as the process already demands significant time per reviewer. We found little evidence that award-winning papers are less costly reproduce than average papers. As awards bring a spotlight to selected publications, we strongly recommend strengthening the role of reproducibility in the criteria used for adjudicating awards, to drive community changes through highly visible examples of best practices and increase the effectiveness of these measures. Furthermore, venues could supplement the review process by assigning a 'reproducibility reviewer', e.g. as done by the AutoML conference. This can distribute the workload and bring a dedicated reproducibility focus to the review process and thus stronger enforcement of standards. Alternatively, venues may include a 'reproducibility papers' track, as, found in e.g. ACM MM or ECIR. Another stimulant for reproducibility is the use of workshops and challenges such as the Reproducible AI workshop or the Machine Learning Reproducibility Challenge. Hosting such tracks, workshops or challenges place a highlight on reproducibility and enables publication of reproduction studies.

We hypothesised that studies published at venues with stricter requirements will be less costly to reproduce. We only partially accept this hypothesis; although venues with stricter requirements generally cost less, we find more factors that play an important role to whether strict requirements are effective; community support and stimulation are important factors whether these standards affect documentation quality. One requirement that may prove effective independently, is requiring authors to submit their implementation as supplementary material rather than only providing an external link; archiving could reduce the erroneous implementation link rate substantially. Although we are not out of the woods yet of the reproducibility crisis, we believe that, with a better, data-driven, understanding of the effectiveness of reproducibility mechanisms we can achieve positive structural, procedural and community changes to mitigate this crisis.

## 8 ETHICS STATEMENT

We acknowledge and confirm that we adhere to the ICLR Code of Ethics. The data collection done for this study was done by the authors, and is sourced from publicly available data. We foresee no negative ethical impacts of this work for the scientific community nor the general public.

## 9 REPRODUCIBILITY STATEMENT

We document the methodology used to produce our dataset in section 4 and Appendix A. The full guidelines, code and dataset can be found in the supplementary material and our GitHub repository [3]. This includes our code used to create our tables and figures for analysis. The few parameters we use, such as the amount of re-samplings in the permutation tests, are documented throughout our paper and are included in our code notebooks as well. We used a standard significance level of $0.05$ for all statistical tests.

---

[3]REDACTED-FOR-ANONYMITY

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

# A   METHOD

In Table 1 we show an overview of features to look for per dimension and the possible impact of increase in cost. For the full guidelines, we refer to the PDF in the supplementary material due to its size. The guidelines consist of 11 pages, including two example reviews with snippets of actual papers as examples, and explain numerous possibilities for which the cost can be increased in bullet points with phrases the reviewer may be looking for in the study. We noted one theoretical example where the Implementation cost could exceed ten by our guidelines; we have not seen any instance where this occurred. These guidelines were used as instructions for the second review as well, and any questions regarding these are logged in the supplementary materials. No questions were answered regarding the paper the second reviewers were reviewing, to avoid biasing their responses.

During the study, we found that the data dimension has somewhat of an 'artificial inflation' when the authors used popular or 'standard' datasets, such as MNIST LeCun et al. (1998) or CIFAR Krizhevsky & Hinton (2009), where the authors did not feel the need to report detailed information on them, especially when it does not play a central role to the method. We decided against making exceptions for popular or common datasets as we found it too complicated to objectively decide what makes a dataset common or 'popular', especially when dealing with smaller subfields that may use less well known datasets to the broader community, but may be considered common in their respective subfields. Thus we did not adopt a separate procedure for these datasets. This also indicates the authors did not provide a (detailed) motivation for choosing a standard dataset for a given task.

We also note in Table 2 that in a few cases, the number of papers slightly varied per venue or year. This we accounted for due to one conflict of interest detected post-review, and the others due to a sampling incident regarding pre-selecting the award-winning papers, resulting in more reviews.

For the statistical analysis we considered many statistical tests to determine the significance between distributions, which we would like to clarify. We only demonstrated the tests over the averaged dimensions, and we would like to elaborate our insights on a dimension/venue-level outcome and update our manuscript accordingly. Normality test showed that the data is not normally distributed ($p < 0.01$ for all). Using Brown-Forsythe test, with the venues as groups, we find that implementation and exp. procedure have strong evidence of nonhomogeneity ($p < 0.01$), significant evidence for configuration (0.045) but not for data (0.30). We do not apply ANOVA as we violate two assumptions: Normality and homogeneity. We applied the Kruskal-Wallis test instead: For each dimension, there is a statistical significant difference for at least one group ($p < 0.003$ for all). We considered the following tests for comparing venues per dimension:

- Kolmogorov-Smirnov: Non-parametric test, does not assume normality distribution. However, it does not have a one sided version.
- Friedman: Requires repeated measures which we do not have.
- Wilcoxon signed rank: An alternative to the t-test when normality cannot be assumed, but requires paired observations
- Mann-Whitney U / Wilcoxon rank-sum: No normality assumption, does not require same size samples or paired data but assumes equal variance
- Permutation: No assumption about normal distribution or variances. For robustness, we increased resamplings to 1 000 000

We decided against p-value correction (Bonferonni) to reduce the false positive rate, as our hypothesis is per single test rather than using a family of statistical test for a single hypothesis, and instead opted for interpretation of the results for practical significance.

## A.1   ASSUMPTIONS ON THE ENTIRE POPULATION

Our analysis is widely conducted to compare venues, and hence our sampling strategy was conducted uniformly at random per venue per year, with the same sample sizes. However, in a few cases, we also presented findings across the entire population; the average public data rate, average implementation link rate and its subsequent error rate. As our sampling strategy does not take into account the proportionality of each venue to the population, we weighted these metrics to the *estimated* rates by determining the proportion of empirical publications of each venue per year and calculating their contribution to the averages accordingly. This resulted in the following value shifts in our study:

- The estimated public data rate, measured in our dataset at $93.95\%$, calculated to $94.01\%$ after weighing the sources per year.

- The estimated public implementation rate, measured in our dataset $62.13\%$, corrected to $70.52\%\%$ after weighing the sources per year. This is the most substantial change, due to the substantially lower rates found in JMLR/JAIR, whose impact is more limited to the overall population than other venues due to a lower amount of publications per year.

- The estimated implementation URL error rate, measured in our dataset at $6.59\%$, corrected to $6.72\%$.

## A.2 SECONDARY REVIEW

A highly important part of this work is the secondary review to determine the reproducibility of our methodology, as described in Section 5.1. There are a few details that we would like to highlight with regards to these results. We have acquired our secondary reviewers through our research network, where doctoral candidates, post-docs and professors were considered for their proficiency. To ensure a diverse group of secondary reviewers, we used various channels with a diverse background. For example, the research directions of our network varies strongly. Among the secondary reviewers, we recorded the following primary research directions: Reinforcement Learning, Neural Network Verification, Music Information Retrieval, Robustness, Algorithm Selection/Configuration, Multi-Agent systems, Time series analysis, SAT solving, Hyperparameter Optimisation, Data Mining, and, ML for Earth Observation. Thus, the communities they are part of which impact their views and judgements are diverse. This was done to reflect a wide variety of researchers from the field of AI, and mitigate possible biases in this part of the dataset. We subsampled our dataset using the same strategy as our initial sampling, and ensured each venue and year received a proportional amount of reviews to be representative of the distribution of the dataset.

Another important subject of this analysis is the sample size; we acquired 46 secondary reviews for our entire dataset ($5.01\%$). Although larger sample sizes are always highly desirable in any data-driven study, we faced resource constraints due to the complexity of the task; we had strong requirements for the academic expertise of the reviewers, and conducting a review is time consuming. We do find the sample size is sufficient for statistically robust claims based on the central limit theorem, and hence do not expect this outcome to change substantially with a larger sample size under this strategy. This leads us to another important point of our method; the acquisition of reviewers was that this was done through our network. Although diverse, as discussed before, an inherent potential bias remains in place in contrast to sampling reviewers uniformly at random on the entire population of proficient reviewers.

This is a common subject in studies containing human subject such as medical or social sciences, where a form of participation bias is often a concern (Elston, 2021; Lesser et al., 2023). We have mitigated this where possible, by providing the secondary reviewers with a neutral written document to use for the evaluation. Furthermore, we instructed the participants that no questions will be answered on how the evaluation should be conducted and that their decision should follow from their understanding of the guidelines. Ideally, we would be able to randomly sample proficient reviewers from the entire population. However, if we were able to easily contact these individuals at a large, non-biased, scale, a non-participation bias may occur; by relying on more passive recruitment they introduce other forms of participant bias. This is likely to lead to only those responding who are more interested in the subject or are connected with the organiser for example, which could lead us back introducing new or similar biases. Hence, although there may be limitations to our methodology to consider, we have mitigated them to the best of our ability when possible.

## A.3 CORRELATING COST TO SUCCESS

Our evaluation has been primarily focused on the quality of documentation and its expected impact on unnecessary labour. We have aimed to create a robust and reproducible methodology as we evaluated in Section 5.1. To extend upon this, we also investigate how our estimated costs directly correlate with the reproduction of a study. As these reproductions are costly, we rely on the dataset of Gundersen et al. (2025), where the authors published the success rate of reproducing studies using public data within a 40 working hour time limit. There are a several biases in this dataset that should be noted:

|  | % Exp. Reproduced | % Exp. Completed | Success |
|---|---|---|---|
| Implementation | **-0.37** (0.03) | **-0.31** (0.03) | **-0.37** (0.02) |
| Data | -0.01 (0.49) | -0.08 (0.32) | 0.07 (0.65) |
| Configuration | **-0.32** (0.03) | **-0.38** (0.01) | **-0.32** (0.05 [a]) |
| Experimental Procedure | -0.16 (0.19) | -0.18 (0.16) | -0.12 (0.28) |

Table 4: The Kendall correlation coefficient between our dimensions and the recorded reproduction success on the dataset from Gundersen et al. (2025). The first column denotes how many experiments of the original authors were successfully reproduced (identical or consistent results), the second column how many in experiments were completed in total (including inconsistent results) and whether the attempt was considered successful ('S' or 'PS') or not ('NR' or 'F'). The P-values of each correlation are placed in brackets, significant results are marked in **bold**.

[a] The measured P-value was 0.0467, rounded to 0.05 in the table.

- The authors use publications from a different time frame (2011 to 2016). Standards of documentation have changed over the years, which may bias the data.

- The authors use *only* publications that use public data for their experiments.

- The authors select the most highly cited publications from their time frame.

- The authors use publications from other venues than those included in this study. Although our method is generalised towards all empirical AI studies, this may imply a shift in population as the community (standards) of researchers from these venues may vary from those evaluated in this work.

- The authors have decided on a hard cut off of forty working hours per study. Working hours also included computational time; this is not considered labour by the definition of reproducibility cost in our work.

A straightforward approach would be to correlate our cost with is the recorded working hours per attempt. However, the dataset is relatively small, and unsupportive in this regard; in two cases, the authors determined no results could be acquired before the time limit due to lacking information, thus limiting the expressiveness of the recorded labour hours of the quality of documentation. Secondly, the majority of the attempts (14 out of 22) ran into the time limit, both in success and failures, making the variable rather constant. Thus, we repose our question towards the data as follows; how strong are our costs correlated with reproduction success within a forty hour working limit? We hypothesise that higher costs imply lower success rates.

We apply the Kendall correlation coefficient to determine the relation between our dimensions and the success rates of the study. We want to analyse the amount of experiments successfully reproduced, the amount of experiments performed overall, and whether the study was considered successfully reproduced or not. Note that we also consider the *amount* of experiments completed, regardless of the outcome; we consider in our method the presence and quality of documentation, but do not claim to verify its contents. Hence, the amount of experiments completed is still a form of success (i.e. the authors were able to conduct an experiment) even if the result was inconclusive. This metric is of course less positive than the other success metrics. The results are found in Table 4. Based on our hypothesis we expect an inverse negative correlation; the higher our cost, the lower the reproducibility. In the table we can see that across the board, the dimensions show moderate correlations with the three properties, except for the data dimension. Based on the previously discussed biases, this is easily explainable; the data dimension has an overall low score due to the authors selecting studies with only publicly available data, which generally implies a low cost by our methodology. Thus this variable is less insightful or predictive of successful reproduction attempts in this dataset. Although we could have included eight samples of their dataset which use private data, studies 22-30, we felt this is misrepresentative as a reproduction attempt was not even conducted. Implementation and Configuration dimension have moderate and significant correlations for all three metrics; Implementation has a slightly stronger correlation for reproduction and successful outcome, whereas Configuration has a slightly stronger correlation with regards to whether an experiments was conducted.

The Experimental Procedure has no significant correlations for any of the measurements. This is can partially be explained by the fact that overall the studies scored quite well in this dimension, similar to our original study (Highest recorded value was four in this subset). We did not expect a single dimension to have a 'near-perfect' correlation with success; the lack of documentation can occur in any dimension and the dimensions may have a joint impact on whether a study can successfully be reproduced or not. Furthermore, upon investigating the problems that occurred during the study, we find that very few problem types are (partially) related to the Experimental Procedure (P1, P4, P9, P10, P13) and many reproduction attempts that encountered such a problem still resulted in a success. These problem types have a strong relation with our work and can give concrete insights into the limitations of our method.

**Limitations of our method**    The authors measured problem types P1, ..., P20 per reproduction attempt, but not present per study in the original publication. We requested the data regarding which study encountered which problem type, which they kindly provided us with. The authors encountered twenty different problems, the majority of which is covered in the documentation evaluation of our method; in fifteen cases we can determine that the subject is covered in our assessment of at least one dimension. However, five of these cases (P1, P7, P14, P19 and P20) are not covered, which amounts to $18.84\%$ of the encountered issues. This is a limitation of our method (i.e. encountered documentation issues that are not captured by our methodology), that we would like to discuss.

- (P1) "Method code is shared, but not experiment code.": our methodology focuses on the documentation of the implementation of the newly presented method. The authors find this issue in five attempts; only one occurred in an unsuccessful attempt.
- (P7) "Random seeds and random number generators not specified.": we explicitly decided against including such values into the cost assessment, as we believe that outcomes should generally be statistically robust and reproducible under different stochastic circumstances. The authors find this issue in four attempts, none of which resulted in a failed attempt. Hence we conclude that not including this into our measurement is not a limitation, but rather that our choice is supported by the (limit) data set.
- (P14) "The article contains an error.": this is a very important problem, that can only be detected upon deeper inspection (and perhaps actual reproduction attempt) of a study. The authors encountered this once, but the attempt was still successful.
- (P19) "Results are presented in a way that makes a comparison hard.": this problem is focused on the presentation of the method. This is a very important topic that is hard to objectively capture. The authors encountered this issue twice, one of which turned out to be unsuccessful.
- (P20) "Lack of access to hardware or software needed to conduct the experiment.": this problem is not related to the documentation of the authors, but rather the accessibility of resources and thus not included in our method based on our definitions in section 4.

## B    EXTENDED BACKGROUND

In this work, we use the term 'reproducibility' to refer to a general scientific concept, rather than limiting it to the field of AI or computer science, for which we find two possible standards: Goodman et al. (2016) and Gundersen (2021), who both use quite similar definitions and terminology. Goodman et al. (2016) separates the definitions of reproducibility into three types: methods, results and inferential reproducibility, which partially overlap with the definitions in Gundersen (2021), who define three degrees of reproducibility: outcome, analysis and interpretation reproducible. Both Goodman et al. (2016) and Gundersen (2021) present their types of reproducibility within a hierarchical ordering. Goodman et al. (2016) refer to methods reproducibility as the ability to repeat the procedure of the study on the level of implementation, e.g. "methodologically reproducible", where the experimental procedure can be repeated using the documentation provided. This does not include the outcome or the conclusions that are drawn from the experiment. There is a discrepancy between fields here: in psychology, methodologically reproducible would refer to mainly reproducing an experimental set-up, whereas in computer science, this would also refer to the reproducibility of the implementation or execution, e.g. code. Although the other categories and degrees, such as results reproducibility

and outcome reproducibility or inferential and interpretation reproducibility, are more overlapping between these studies, for conciseness and clarity, we refrain from using definitions of categories and degrees. We find, however, the definitions in Goodman et al. (2016) to be more ambiguous, as they lack any definitions of the independent investigators (the 'reproducers'), as well as the concept of what constitutes the documentation of the method. We also disagree with the reasoning that the search for 'truth' is the main motive for reproducibility. We find our answer in Popper (1934) instead: we argue it is the credibility of a method.

In Crook et al. (2013), the authors discuss various approaches regarding reproducibility in computational neuroscience. The authors define reproduction and replication of an experiment: An independent investigator either reproduces an experiment, or replicates it 'using the same code', and they consider the latter to be 'certainly easier than independent reproduction'. Later the authors redefine these terms as internal and external replicability, and introduce the concept of cross-replicability: The ability to simulate the same model with different software. Lastly, they define reproducibility as the ability to implement a model independently without using the source code of the original authors.

We refrain from using these definitions due to their ambiguity, and limitations. The definition of reproducibility excludes the source code as part of the documentation, which we find an unnecessary constraint. The terms of internal and external replicability are in our point of view redundant: Reproducibility should be measured based on the documentation the authors provide to the world, and should thus not vary based on the relation to the independent investigator and one could even argue that a closer relation could create a conflict of interest. The information provided to a reader for 'external replicability' should be the same as for reproducibility. Lastly, the term Cross-replicability refers to the ability to re-implement a method outside the domain of the source code. We find this a constrained version of the re-implementation of methods within reproducibility.

In Poldrack (2019), the reproducibility cost in neuroscience is discussed, where the author highlights the gap between early career researchers, and their desire to apply the best practices, that may conflict with their career goals. The author outlines the problems with the lack of resources and advises the students to pivot their research questions so that the resources allow them to answer or rely on publicly available or shared data to achieve a sufficient sample size. However, the author also illustrates that the incentive for early career researchers to apply best practices harms their chances in the short-term job market, as the investment could lead to them acquiring fewer publications overall. The author addresses the need for senior researchers to stimulate better science overall instead of focusing on the need to find, for example, positive outcomes for every study. We believe that, although some of these issues within the field of neuroscience do not directly translate into the field of Artificial Intelligence, the key actors such as the need for resources, publicly available data and the mentality of "Publish or Perish" that can be extremely detrimental to the quality of studies and subsequent reproducibility.

## C    EXTENDED QUANTIFICATION STUDIES

In Raff (2019), the authors define features to test for correlation with reproducibility of studies. The most directly linked feature to our work is the specification of hyperparameters, which we evaluate under configuration cost, and pseudo code, which we evaluate under implementation cost. It is important to note that both of these features are considered in Raff (2019) as 'mildly subjective', which also finds its place in our work as we enable our reviews to diverge from the set guidelines. One significant feature that we do not include in our work is required compute. We have chosen to exclude this, as our focus lies on the documentation provided by authors rather than the accessibility of compute. Another reason, unrelated to the work of Raff (2019), is that the variation that can be caused by different low-level systems (such as a CPU/GPU architecture or drivers) is more closely aligned with *outcome* reproducible rather than *interpretation* reproducible, e.g. being able to draw the same conclusion. Variations in (low-level) system software may impact the results outcome in unexpected ways, even if factors of stochasticity are well documented; for example seeds for non-deterministic algorithms. However, exactly reproducing the outcomes is a highly strict view on reproducibility which often may be unnecessary; as long as outcomes support the same interpretations for the hypothesis (for example, varying outcomes but statistical tests still prove the same significances) we find this sufficient to determine a study reproducible.

Due to its widespread applicability and popularity, the impact of irreproducible machine-learning research is not limited to its field Ball (2023); Gibney (2022); La Malfa et al. (2024). In Kapoor

& Narayanan (2023), the authors study 642 publications across 17 scientific fields, ranging from medicine, satellite imaging and cybersecurity to the political sciences. They identified 294 cases of data leakage and classified them into eight categories. These classifications cover situations such as ill-separated training and test sets, or the lack thereof entirely, as well as more complicated examples such as temporal leakage or feature selection and dataset pre-processing on the test set. The authors conclude that the application of ML methods has many pitfalls and that each is rediscovered independently by each community. To prevent or mitigate these issues, Kapoor et al. (2024) provides recommendations for a stronger review process regarding reproducibility documentation, which can help to alert the authors to their mistakes, thus contributing to more realistic or well-founded outcomes. The authors conclude that this identification process of these pitfalls is a first step in the direction of addressing this interdisciplinary crisis. Secondly, increasing the focus on documentation on the application of AI methods regarding reproducibility emphasises the importance to the community.

## D EXTENDED FIGURES AND TABLES

In this section, we present several graphs and tables that are used as a supporting role for our main results. In Table 5 we present the median, quantiles and data skew. In Table 6 we represent the results of the normality test. In Table 7 we show the permutation test between all sources per dimension, testing which sources are significantly less costly than others. In Table 8 we present the results of the Kolmogorov-Smirnov test and permutation test for award winning papers. In Figure 2 we present the correlation per source and overall.

| **Implementation** | | | | | | **Data** | | | | |
|---|---|---|---|---|---|---|---|---|---|---|
| | Q1 | Median | Q3 | IQR | Skew | Q1 | Median | Q3 | IQR | Skew |
| AAAI | 2 | 6.50 | 9 | 7 | -0.09 | 1 | 2.00 | 3 | 2 | 1.33 |
| IJCAI | 2 | 5.00 | 9 | 7 | 0.14 | 1 | 2.00 | 4 | 3 | 1.47 |
| ICLR | 2 | 2.00 | 8 | 6 | 0.86 | 2 | 3.00 | 4 | 2 | 1.31 |
| ICML | 1 | 2.00 | 8 | 7 | 0.72 | 1 | 3.00 | 3 | 2 | 0.79 |
| NeurIPS | 2 | 2.00 | 5 | 3 | 1.16 | 2 | 3.00 | 4 | 2 | 0.92 |
| JAIR | 2 | 3.00 | 9 | 7 | 0.32 | 2 | 3.00 | 4 | 2 | 0.85 |
| JMLR | 2 | 3.00 | 10 | 8 | 0.17 | 2 | 3.00 | 4 | 2 | 0.44 |
| **Configuration** | | | | | | **Experimental Procedure** | | | | |
| | Q1 | Median | Q3 | IQR | Skew | Q1 | Median | Q3 | IQR | Skew |
| AAAI | 2 | 3.50 | 5 | 3 | 0.94 | 1 | 2.00 | 3 | 2 | 1.39 |
| IJCAI | 2 | 3.00 | 5 | 3 | 1.04 | 1 | 2.00 | 3 | 2 | 1.60 |
| ICLR | 2 | 3.00 | 4 | 2 | 1.64 | 1 | 2.00 | 3 | 2 | 1.32 |
| ICML | 1 | 3.00 | 3 | 2 | 1.73 | 1 | 1.00 | 3 | 2 | 1.04 |
| NeurIPS | 3 | 3.00 | 4 | 1 | 1.36 | 1 | 2.00 | 3 | 2 | 0.85 |
| JAIR | 1 | 2.50 | 4 | 3 | 1.35 | 1 | 1.00 | 2 | 1 | 1.68 |
| JMLR | 1 | 3.00 | 4 | 3 | 1.18 | 1 | 1.00 | 2 | 1 | 1.80 |

Table 5: The median, first and third quantile and interquantile range of each dimension per venue. Visualisation of the entire dataset can be found in Figure 1. The skewness of the distribution is nearly always positive, with the exception of AAAI in the implementation dimension, thus motivating the representation of median rather than mean as it is frequently substantially affected by right skewed distribution outliers except for the implementation dimension which has relatively mild skew values.

## E DATA COLLECTION

In the sampling process, we manually excluded the following categories of papers from each source, as they generally lack a full empirical evaluation:

- In AAAI we exempted the (student) abstracts

|  | Implementation | Data | Configuration | Experimental Procedure |
|---|---|---|---|---|
| AAAI | 1728.65 (0.00) | 35.00 (0.00) | 17.19 (0.00) | 36.18 (0.00) |
| IJCAI | 1398.58 (0.00) | 36.10 (0.00) | 18.00 (0.00) | 43.33 (0.00) |
| ICLR | 29.38 (0.00) | 37.34 (0.00) | 51.78 (0.00) | 34.83 (0.00) |
| ICML | 72.84 (0.00) | 12.35 (0.00) | 52.38 (0.00) | 18.42 (0.00) |
| NeurIPS | 20.97 (0.00) | 22.03 (0.00) | 34.03 (0.00) | 12.82 (0.00) |
| JAIR | 3948.80 (0.00) | 10.70 (0.00) | 23.84 (0.00) | 39.46 (0.00) |
| JMLR | 1075.33 (0.00) | 8.91 (0.01) | 21.25 (0.00) | 48.88 (0.00) |

Table 6: Normality test (D'agostino & Pearson, 1973) of all venues and dimensions, where the p-values are shown in brackets. The results are statistically significant for each venue. Thus, we find evidence for each venue that the results are not normally distributed.

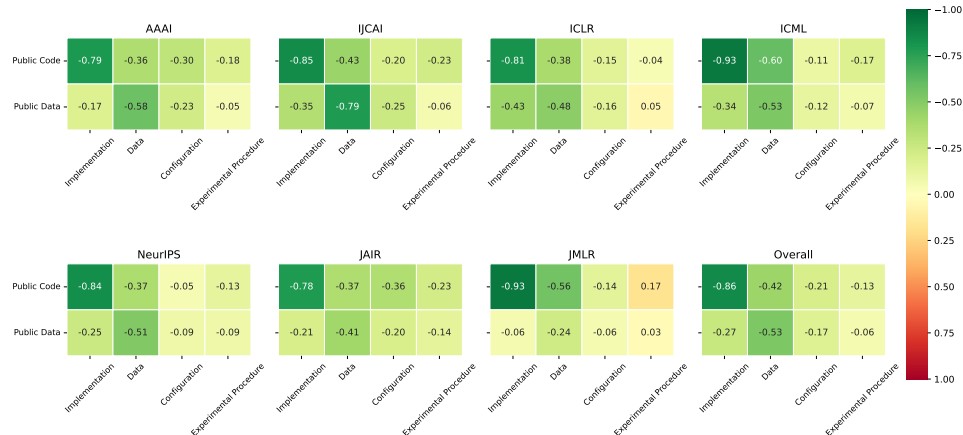

Figure 2: Kendall Correlation heat map of the dimensions and the collected metrics, namely implementation link (URL) and proportion of public data sets (Public Data). A negative correlation indicates lower cost.

- In IJCAI we exempted the survey, doctoral consortium, and early career tracks. The Special Track on AI for Good (Projects) and 'demonstration track' were curated for empirical evaluations. 'Extended abstract' papers were included when the full version was linked.
- In ICLR, ICML and NeurIPS we exempted papers originating from a journal.
- In ICML we curated Position Papers for an empirical evaluation.

In our data scraping, we supplemented the data from OpenReview manually to ensure representative documentation from the point of view of a reviewer for each paper. We have two specific notes: JMLR 2024 only had 49 accepted papers at the time of sampling, thus all were used. In JAIR 2022 "Marginal Distance and Hilbert-Schmidt Covariances-Based Independence Tests for Multivariate Functional Data" where code is available upon request. We requested, received, and reviewed accordingly. Regarding the reviews of papers with awards, we included from each source/year the papers with 'best paper award' up to 8 papers. If more were presented with such an award 8 were sampled at random due to resource and time constraints and to avoid data undersampling of the 'regular' paper population.

During sampling, the following papers were rejected based on possible conflict of interest: *Redacted due to anonymity*

**Implementation**

|  | AAAI | IJCAI | ICLR | ICML | NeurIPS | JAIR | JMLR |
|---|---|---|---|---|---|---|---|
| AAAI | - | 0.83 | 1.00 | 1.00 | 1.00 | 0.91 | 0.77 |
| IJCAI | 0.18 | - | 1.00 | 0.99 | 1.00 | 0.67 | 0.45 |
| ICLR | **0.00** (-1.67) | **0.00** (-1.25) | - | 0.34 | 0.62 | **0.01** (-1.04) | **0.00** (-1.33) |
| ICML | **0.00** (-1.48) | **0.01** (-1.07) | 0.67 | - | 0.77 | **0.04** (-0.86) | **0.01** (-1.14) |
| NeurIPS | **0.00** (-1.78) | **0.00** (-1.37) | 0.40 | 0.25 | - | **0.01** (-1.16) | **0.00** (-1.44) |
| JAIR | 0.09 | 0.34 | 0.99 | 0.97 | 0.99 | - | 0.29 |
| JMLR | 0.24 | 0.57 | 1.00 | 0.99 | 1.00 | 0.72 | - |

**Data**

|  | AAAI | IJCAI | ICLR | ICML | NeurIPS | JAIR | JMLR |
|---|---|---|---|---|---|---|---|
| AAAI | - | 0.15 | 0.11 | 0.30 | **0.02** (-0.41) | 0.15 | **0.00** (-0.67) |
| IJCAI | 0.86 | - | 0.49 | 0.74 | 0.25 | 0.53 | **0.05** (-0.42) |
| ICLR | 0.91 | 0.54 | - | 0.78 | 0.24 | 0.56 | **0.03** (-0.41) |
| ICML | 0.73 | 0.29 | 0.24 | - | 0.07 | 0.29 | **0.00** (-0.56) |
| NeurIPS | 0.98 | 0.77 | 0.78 | 0.94 | - | 0.81 | 0.11 |
| JAIR | 0.87 | 0.50 | 0.47 | 0.74 | 0.22 | - | **0.03** (-0.44) |
| JMLR | 1.00 | 0.96 | 0.97 | 1.00 | 0.90 | 0.98 | - |

**Configuration**

|  | AAAI | IJCAI | ICLR | ICML | NeurIPS | JAIR | JMLR |
|---|---|---|---|---|---|---|---|
| AAAI | - | 0.75 | 1.00 | 1.00 | 0.95 | 1.00 | 0.97 |
| IJCAI | 0.27 | - | 1.00 | 1.00 | 0.81 | 0.97 | 0.86 |
| ICLR | **0.00** (-1.10) | **0.00** (-0.90) | - | 0.55 | **0.00** (-0.64) | 0.20 | **0.04** (-0.53) |
| ICML | **0.00** (-1.12) | **0.00** (-0.92) | 0.48 | - | **0.01** (-0.66) | 0.19 | **0.03** (-0.55) |
| NeurIPS | 0.05 | 0.20 | 1.00 | 1.00 | - | 0.89 | 0.65 |
| JAIR | **0.01** (-0.84) | **0.04** (-0.64) | 0.82 | 0.82 | 0.12 | - | 0.23 |
| JMLR | **0.04** (-0.57) | 0.15 | 0.97 | 0.97 | 0.37 | 0.78 | - |

**Experimental Procedure**

|  | AAAI | IJCAI | ICLR | ICML | NeurIPS | JAIR | JMLR |
|---|---|---|---|---|---|---|---|
| AAAI | - | 0.86 | 0.97 | 1.00 | 0.98 | 1.00 | 1.00 |
| IJCAI | 0.16 | - | 0.80 | 0.98 | 0.84 | 1.00 | 1.00 |
| ICLR | **0.03** (-0.31) | 0.23 | - | 0.91 | 0.58 | 0.98 | 1.00 |
| ICML | **0.00** (-0.49) | **0.02** (-0.30) | 0.11 | - | 0.13 | 0.82 | 0.93 |
| NeurIPS | **0.02** (-0.33) | 0.18 | 0.47 | 0.90 | - | 0.98 | 0.99 |
| JAIR | **0.00** (-0.60) | **0.01** (-0.42) | **0.03** (-0.29) | 0.22 | **0.03** (-0.27) | - | 0.70 |
| JMLR | **0.00** (-0.66) | **0.00** (-0.48) | **0.01** (-0.35) | 0.09 | **0.01** (-0.33) | 0.35 | - |

Table 7: Permutation test of 1 000 000 resamples between sources per dimension, where each row is tested to be smaller than each column. The table presents the found $p$-values per comparison/test. The statistically significant results are highlighted in **bold** with the statistic values (distances) in brackets.

| Implementation | | | | Data | | |
|---|---|---|---|---|---|---|
| | KS-Test | P-Test | | | KS-Test | P-Test |
| AAAI | 0.20 (0.99) | 0.48 (0.64) | | AAAI | 0.14 (1.00) | 0.07 (0.55) |
| IJCAI | 0.35 (0.30) | 2.33 (0.95) | | IJCAI | 0.14 (0.99) | 0.39 (0.70) |
| ICLR | 0.14 (0.94) | -0.05 (0.48) | | ICLR | 0.14 (0.93) | -0.24 (0.33) |
| ICML | 0.18 (0.51) | -1.09 (0.11) | | ICML | **0.33** (0.03) | **-1.12** (0.00) |
| NeurIPS | 0.15 (0.81) | -0.41 (0.31) | | NeurIPS | 0.22 (0.34) | -0.65 (0.06) |

| Configuration | | | | Exp. Proc. | | |
|---|---|---|---|---|---|---|
| | KS-Test | P-Test | | | KS-Test | P-Test |
| AAAI | 0.28 (0.86) | -0.72 (0.30) | | AAAI | 0.43 (0.36) | -0.51 (0.28) |
| IJCAI | **0.64** (0.00) | 2.49 (1.00) | | IJCAI | 0.26 (0.68) | 0.63 (0.93) |
| ICLR | 0.15 (0.87) | -0.44 (0.19) | | ICLR | 0.15 (0.87) | -0.24 (0.26) |
| ICML | 0.12 (0.93) | -0.57 (0.14) | | ICML | 0.09 (0.99) | -0.34 (0.11) |
| NeurIPS | 0.20 (0.44) | 1.06 (0.97) | | NeurIPS | 0.10 (0.99) | -0.25 (0.20) |

Table 8: Kolmogorov–Smirnov test (Dodge, 2008) and Permutation test per source and dimension between papers with and without awards. The P-values are in brackets. The statistically significant results are highlighted in **bold**.