# OpenReview forum: "The Cost of Reproducibility in Artificial Intelligence"
_ICLR.cc/2026/Conference — Submitted to ICLR 2026_

### Official Review · Reviewer_wy9e · 2025-10-18

**Soundness:** 3
**Presentation:** 4
**Contribution:** 2
**Rating:** 6
**Confidence:** 3

**Summary:**

This paper introduces a rubric-based quantification of “cost of reproducibility” that operationalizes documentation quality across five dimensions—implementation, data, configuration, experimental procedure, and expertise—each scored from 1 to 10 via additive penalties. The authors sample about a thousand papers from seven leading ML/AI venues, restrict evidence to materials directly linked from the publication, and analyze venue-level differences using nonparametric tests. A small reliability study with 46 double-annotated papers reports excellent ICCs for implementation, data, and configuration; good for experimental procedure; and only moderate for expertise, after which the expertise dimension is excluded from downstream analyses. Key results include that ML conferences exhibit significantly lower implementation cost than AI conferences and journals, that configuration cost is lowest at ICLR/ICML despite NeurIPS’ stricter checklist enforcement, and that award-winning papers are not systematically less costly to reproduce. The paper argues that community norms and enforcement during review are more predictive of documentation quality than the mere presence of checklists or appendices.

**Strengths:**

## Novelty and contribution
The paper provides a concrete and transparent operationalization of reproducibility “cost” as documentation shortfalls rather than computational resource requirements. The five-dimensional rubric in Table 3 is detailed, public, and anchored in practical indicators. By sampling across top venues and both conferences and journals over multiple years, the work delivers a cross-sectional audit of current practice that goes beyond prior smaller-scope studies and moves from binary reproducible/irreproducible labels to a graded notion of documentation burden.

Interesting findings are e.g. the differences between major ML and AI conferences, and the fact that award papers are not easier to reproduce on average.

## Correctness
The methodological stance is clear and internally consistent: evidence is limited to materials directly linked by the publication, thereby standardizing what a diligent reader would find. Reliability is explicitly addressed; while the double-annotation is small, the ICCs for the retained dimensions are excellent to good, supporting use of those scores in venue comparisons.

It is especially commendable that the authors report distributional diagnostics and select nonparametric tests accordingly. Apart from reproducibility, parts of the field are also struggling with measures of statistical significance. It is pleasant to see that a paper analyzing the reproducibility crisis does so with due diligence on the statistical side.

## Presentation and clarity
The manuscript is carefully organized, with figures and tables that map directly to claims in the text. The contrast between venues is repeatedly tied to concrete quantities, and deviations (e.g. the slightly reduced sample size through a sampling mistake) are openly discussed.

**Weaknesses:**

## Novelty
While the cost is novel, it is also mostly a heuristic without too much justification. In addition, the paper's main findings do not directly result in novel recommendations for measures that venues could implement to strengthen reproducibility. The recommendations in the conclusion are fairly high-level and could have been proposed independently of the study.

## Cost design
The primary limitation is the rater design. A single rater annotated all papers, and only 46/918 were double-annotated, selected by volunteers from the authors’ network and by interest. This introduces potential selection bias and may overestimate reliability. The moderate ICC for expertise led to dropping that dimension, but it also means that the rubric initially included a dimension that was not reliably operationalized.

The score also oversimplifies things. Depending on the paper, it is not clear how important a dataset is compared to the code. In addition, the implementation may often cover the configuration and/or data. Thus, treating these categories as separate dimensions may be problematic. A single score will always contain an implicit weighting of categories. This becomes especially problematic since the penalties from Table 3. are pretty arbitrary in their weighting.

Within the rubric, equal weighting of datasets in the data dimension can penalize papers that include many secondary benchmarks while emphasizing a primary dataset. The decision not to treat widely used benchmarks differently may improve objectivity, but may also disadvantage subfields with many commonly used datasets.

## Presentation and clarity
Several central elements are relegated to the appendix, which makes readability a bit harder. Table 3 is essential for understanding scores and should be summarized more prominently in the main text. This is most likely a space concern and could be fixed with the extended 10 page limit for the rebuttal version. Minor inconsistencies (93.95% vs 93.85% public data rate), phrasing (line 376, “implementation links occurs”), and formatting glitches (citation formatting around line 881) should be corrected.

Scope and granularity
The study aggregates across subfields and years despite policy changes within the window (e.g., NeurIPS 2024 checklist mandate versus ICLR’s optional statement). Year-wise trends could strengthen causal narratives around policy impacts. Similarly, stratification by subfield (e.g., vision, NLP, RL) could reveal domain-specific documentation norms that confound venue-level comparisons. The SCOPUS enrichment in Appendix D.1 is peripheral and thinly analyzed; if kept, it would benefit from reporting effect sizes and uncertainty or else be moved fully to supplementary material.

**Questions:**

- Though usually (rightfully!) discouraged for review, it would be interesting to compute the correlation of the authors judgment with several of the strongest LLMs. Have you checked your score against one of those? If successfully cross-checked on the main data from the paper, this could be used to check thousands of papers without much effort.

- Have you considered the interplay of your categories, e.g. implementation with configuration? They are presumably not independent from each other.

- How sensitive is your cost against changes to the penalties in Table 3?

- Have you made experiments on how strongly the cost is correlated with the success rate of actually reimplementing the papers? E.g. in the spirit of Gundersen et al. from 2025, have you tried to implement some of the papers and correlate that with the cost? Can you maybe reuse the results of Gundersen by also rating those papers according to your metric?

---

> ### Author Response · Authors · 2025-11-17
> **Rebuttal [1/3]**
>
> We would like to thank the reviewer for their time and extensive review of our manuscript.
>
> # Novelty
> Although parts of our recommendation could have been made independently of this work, we find that the need for data-driven support is highly important; for example, our hypothesis that stronger requirements by venues results in better documentation was found to be only partially true. We also aimed to avoid making any suggestions that are only partially supported by the data as this would be detrimental to our work. More granular insights and recommendation could however prove very effective with relatively small adaptations for organisers; we have added constructive feedback on how the error rate of broken implementation URLs can be reduced, by requiring it as supplementary material submission for archiving purposes. We have marked this change to the conclusion in red.
>
> # Cost Design
>
> ## Rater design
> The resources required for meta-studies on reproducibility is an often recurring problem in this field; there are large amounts of time investments required for a single study. For example, Gundersen et al. (2025) [1] collectively had to invest up to 880 working hours to acquire their dataset for a sample size of 22, Gundersen & Kjensmo (2017) [2] used a sample size of 400. Raff (2019) [3,4] had a sample size of 255. The most constraining factor for this part of the experiment is that this work can overall not be outsourced and requires highly trained or educated individuals to conduct. With regards to the secondary reviewer; although there are possible sources of biases, we aimed to reduce this as much as possible. We have clarified this in section 5.1. (marked in red), that we would like to summarise here; secondary reviewers were allowed to select papers based on paper titles alone, to match their interest. Although this selection is a bias, we believe it to be a positive one s.t. the reviewed paper matches their expertise. Secondly, secondary reviewers were not allowed to select the same paper, to avoid overrepresentation of a single work in the dataset. Thirdly, the secondary reviewers were only allowed to select a randomly sampled subset to ensure spread across venues and years; each venue received at least five secondary reviews, and each venue year has at least one secondary review.
>
> ## Score
> The relation between the dimensions exists, but we make little assumptions on what this exact relationship is. With regards to the importance of code versus data; note that the implementation dimension focusses on _new_ methods (section 4.1, paragraph 2 "Implementation Cost."). Thus, if a study introduces a new benchmark dataset for example and provides measurements of baselines that do not include a new method introduced in this work, the implementation cost will be overall quite low as less documentation is expected of the authors. The documentation of the baselines is the reponsibility of their respective authors. However, datasets introduced by the authors are evaluated more thoroughly, as independent investigators cannot rely on other works for documentation on how this dataset was acquired for example.
>
> With regards to the overlap between the repository and implementation/data/configuration; we treat supplied code (repositories or supplementary material) as a key part of the documentation provided by the authors. Authors may decide to document information here with regards to all dimensions. The implementation dimension is not reduced to just this subset of documentation; key design choices for the implementation (or for example pseudo code or diagrams) may just as well be present in the main body of the paper or the appendix. The same holds true for data and configuration; we found it often to be the case that authors provide (hyper)parameter tables in the appendix or links to datasets in the body of the paper, as well as placing this information in the repository.
>
> ## Weighting of datasets
> We believe that a clarification is in order here; the datasets were equally weighted by default, but this was left open for the reviewer to adapt accordingly when necessary. We have marked our changes in section 4.2. in red. With regards to widely used datasets; although a (small) penalty may be involved when common datasets are used, we do support our original decision on this matter. One possibility to define 'common' or 'popular' datasets is by citation count of the paper publishing said dataset; we view this as a possible disadvantage to smaller subfields where the amount of citations may be substantially lower yet still be considered common to larger subfield, making it complicated to set a hard threshold. We have clarified this in Appendix A and the caption of table 1 (Formerly table 3 of the appendix) and marked our changes in red.

---

> ### Author Response · Authors · 2025-11-17
> **Rebuttal [2/3]**
>
> # Presentation and Clarity
>
> ## Central elements
> We agree with the reviewer that Table 3 is essential to understand our methodology, and was indeed placed to the appendix due to page limitations. We find this a very important point, and have adapted our manuscript accordingly; Table 3 of the appendix is now Table 1 of the manuscript, and adapted the caption and sentences referring to this table slightly to increase readability. We have also fixed the minor inconsistencies pointed out, and are grateful for such a rigorous review. We have marked the changes in **red** in our manuscript.
>
> ## Scope and granularity
> We agree with the reviewer that a year-on-year impact analysis, or per subfield investigation, could yield insightful trends aside from those already found in the manuscript and were originally intended angles. There are however a few complications that motivated us to refrain from conducting such analyses. Primarily, this is due to the scale of the study; in order to make statistically reliable claims with regards to subfields, a sizeable population must be present for each and the we believe that the current sample size, although relatively large, would need to be expanded to produce any insightful and sound claims. We find a similar situation for year-on-year analyses; as we conducted this study over the past three years, it is difficult to determine any strong changes within the community with regards to documentation over a relatively short timespan. Furthermore, we applied in our method the same sampling sizes per year, per venue, and would for such an analysis require proportional representation of these two dimensions (Number of accepted studies per venue vary strongly, and the amount of publications each year is growing at a rapid pace). Based on these three arguments, we decided to refrain from making claims that our data would likely not or poorly support. This does not mean of course that such analysis would be uninteresting, and originally we had aimed for an even larger population (In terms of venues and amount of years) for this study but were limited in resources.
>
> ### Scopus enrichement
> We have moved the SCOPUS enrichment fully to the supplementary material, as it only serves an interesting angle of analysis for a highly curious reader.

---

> > ### Author Response · Authors · 2025-11-17
> > **Rebuttal [3/3]**
> >
> > # Questions
> >
> > _Though usually (rightfully!) discouraged for review, it would be interesting to compute the correlation of the authors judgment with several of the strongest LLMs. Have you checked your score against one of those? If successfully cross-checked on the main data from the paper, this could be used to check thousands of papers without much effort._
> > We have thus far mainly considered to leaving this for future work; the effort required to conduct this study has been quite large so far, and we felt that, by extending this towards automated evaluation using LLMs, the project would grow quite substantially in workload, especially with the considerations that would be needed to taken into account when verifying the LLM outcomes (e.g. highly likely more manual evaluation before extending this to an even larger scope).
> >
> > _Have you considered the interplay of your categories, e.g. implementation with configuration? They are presumably not independent from each other._
> > Although the dimensions were chosen to be relatively 'orthogonal', i.e. each is aimed to answer a different aspect of documentation, we do not assume them to be independent; studies that choose to release code for example, will often publish parameter configurations alongside it, or the used datasets for example. We attempted to find a correlation between the costs and objective features (open code, open data) in Figure 2 of the appendix. Here we found overall that the presence of code has a substantial impact overall on the three other dimensions, with data being the most affected. This is unsurprising as authors often link or publish datasets with their code. What we did find an interesting outcome, as noted in , is the relatively weak correlation overall between published code and the configuration dimension. We also extended this correlation between dimensions, but found the results less insightful than we had hoped. We can of course include this Kendall Correlation into our appendix, and share the insights and opinions there as well in more detail. Would this in your opinion improve our manuscript?
> >
> > _How sensitive is your cost against changes to the penalties in Table 3?_
> > Table 3 (currently table 1 in the revised version of our manuscript) defines broadly what can be considered a penalty for our measurements. Many guidelines define ranges rather than strict values; we thus reason that our method is more sensitive to restricting changes than those that offer more options as it would forcibly change the awarded penalty, whereas more liberal ranges would have relatively little impact as it would only possibly affect those at the upper limit. Does this answer satisfy your question? If not, could you elaborate on this subject?
> >
> > _Have you made experiments on how strongly the cost is correlated with the success rate of actually reimplementing the papers? E.g. in the spirit of Gundersen et al. from 2025, have you tried to implement some of the papers and correlate that with the cost? Can you maybe reuse the results of Gundersen by also rating those papers according to your metric?_
> > We have considered reviewing the reproduced papers from Gundersen et al. (2025), to determine the correlation of our method and the amount of working hours required to reproduce the method. There were a few biases we considered, mainly that these papers are less representative of the entire population of published research (Only the most highly cited were used) as well as the fact that only studies with public datasets were considered (e.g. ~6.05% of the population is not represented in their dataset). Finally, the papers were selected outside of our time range (2012/2014/2016 vs 2022-2024 in our work). These caveats have thus far moved us to refrain from conducting this analysis, although we do see the apparent value in it. Considering it as an independent experiment to validate our method; it is a relatively simple experiment that we can add to our work. Would such a correlation be a valuable addition to our manuscript i.e. what is your opinion on our listed caveats?
> >
> > # References
> > [1] Gundersen, Odd Erik, et al. "The unreasonable effectiveness of open science in ai: A replication study." Proceedings of the AAAI Conference on Artificial Intelligence. Vol. 39. No. 25. 2025.
> >
> > [2] Gundersen, Odd Erik, and Sigbjørn Kjensmo. "State of the art: Reproducibility in artificial intelligence." Proceedings of the AAAI conference on artificial intelligence. Vol. 32. No. 1. 2018.
> >
> > [3] Raff, Edward. "A step toward quantifying independently reproducible machine learning research." Advances in Neural Information Processing Systems 32 (2019).
> >
> > [4] Raff, E. (2021). Research Reproducibility as a Survival Analysis. Proceedings of the AAAI Conference on Artificial Intelligence, 35(1), 469-478. https://doi.org/10.1609/aaai.v35i1.16124

---

> > > ### Comment · Reviewer_wy9e · 2025-11-23
> > >
> > > Thank you for the thorough response to the review and questions.
> > >
> > > ## Main Concern: Correlation Between the Score and Real Reproducibility
> > >
> > > ### Correlation Between Reviewers
> > > The main issue I see with secondary reviewers is that they were sampled from the *from the authors’ network of researchers*. Since acquaintances are more likely to have correlated views and judgement, bias becomes highly likely. This is especially important because the double review is used to argue that the rest of the papers with a single reviewer are close to the objective score.
> > >
> > > ### Score
> > > You also state that "relation between the dimensions exists, but we make little assumptions on what this exact relationship is". However, if relations do exist, and two columns are closely correlated, then the score may be overly influenced by weighing the same thing effectively twice.
> > >
> > > The ranges for score deductions are also decided ad-hoc as far as I can tell, and the exact value is determined by the reviewer. This undermines the confidence that the score measures what it intends to measure.
> > >
> > > This was also what the question on the sensitivity of the cost was referring to. Is it possible to significantly change the outcomes just by redefining the penalty ranges or (within reason) the rules of the score? If so, how do we know that these rules are sensible? This must somehow be established in my opinion.
> > >
> > > ### Datasets
> > > Thank you for the clarification on the datasets. While your justification makes sense, you also state that "this was left open for the reviewer to adapt accordingly when necessary". So effectively the bias in creating a rule was traded for a bias on the side of the reviewer.
> > >
> > > ### Summary
> > > All points from above raise doubts concerning the validity of the final score for measuring the actual cost of reproducibility. So to answer your final question, yes, I think a comparison to Gundersen et. al would be extremely useful to establish the score as a good measurement of reproducibility, and would significantly strengthen the paper. The one remaining caveat is then the evaluation by a single human reviewer, which could potentially be fixed in a future work by showing that LLM agents can derive highly correlated scores.
> > >
> > > ## Others
> > >
> > > ### Scope and Granularity
> > > The explanation for the omission of a year-by-year or by subfield analysis makes sense. It would be interesting, but I respect the judgement on the statistical significance of the findings, and appreciate that the authors value it over premature results.
> > >
> > > ### LLM-based Analysis
> > > While the time frame of the review period is too tight to conduct the experiments with sufficient rigor, I don't see how the process would be fundamentally different from calculating the ICC for the secondary reviewers. It could be a viable path towards a large-scale study on the raised issue.
> > >
> > > ### Kendall Correlation
> > > I find the correlation interesting, especially since it might lead to recommendations on what categories conferences might want to explicitly demand of authors (e.g. code is sufficient without configs doesn't seem to be a viable policy as per your results). However, this is a minor point. Feel free to include or exclude based on your judgement.

---

> > > > ### Author Response · Authors · 2025-11-26
> > > > **Response to reviewer wy9e**
> > > >
> > > > Thank you for your feedback on these important matters. We have made several changes to our manuscript that we would like to highlight.
> > > >
> > > > # Correlation Between Reviewers
> > > > Although there is a possible source of bias through our network, we would like to clarify that we have taken steps to mitigate these where possible. Our network is highly diverse in terms of research directions, and thus have diverse experiences with various subcommunities within AI (Among others; Reinforcement Learning, Neural Network Verification, Music Information Retrieval, Robustness, Algorithm Selection/Configuration, Multi-Agent systems, Time series analysis, SAT solving, Hyperparameter Optimisation, Earth Observation). We believe this limits this bias, as this entails that they often interact with different communities that impact their views. We have added a section in the appendix (A.2. marked in red) to address this topic based on your feedback and that of reviewer fGe7, as it is of high importance to us that our results are representative and reproducible.
> > > >
> > > > # Score
> > > > Yes, we assume that there is a relation between these columns in the sense that they together may impact how reproducible a study is. However, the dimension were designed to be 'orthogonal' to one another; a subject adressed in one dimension cannot be counted elsewhere. This has been strictly taken careof through the method.
> > > >
> > > > With regards to the score deductions; we noticed that, in an initial trial, authors tend to use various facets of documentation (That substantially contribute to the documentation!) when it comes to the implementation, that we deemed complicated to find a generalisable guideline for. We noticed that we did not clarify this in table 1 and have adapted the manuscript and marked this in red. Thank you for pointing this out to us.
> > > >
> > > > With regards to changing the penalties; by changing the rules or penalties unreasonably one can affect the distribution, but we do not believe that it will significantly impact the _conclusions_ of the paper. By for example reducing or increasing the amount of rules or penalties, the results will be more condensed or finegrained, but our analysis of these outcomes mostly focused on general patterns within the distributions. Hence, we do not think that, within reason, adapting our guidelines will result in significantly different outcomes but perhaps in more finegrained results that could support more detailed conclusions.
> > > >
> > > > # Datasets
> > > > We indeed reasoned that shifting this potential bias allows for more representative evaluation of a study. Under fixed equal weights, datasets used only in supplementary experiments would have the same impact as central datasets, even if they contribute little to the conclusions of the original authors. We reduced this bias by defaulting to the fixed weights i.e. the documentation of the study needs to have some clear indicators to motivate a reviewer to move away from this.
> > > >
> > > > # Comparison to Gundersen et al.
> > > > Thank you for sharing your opinion on this matter. We have dedicated a section in our new version of the manuscript to this subject, see appendix A.3. We would like to summarise our findings briefly here;
> > > >
> > > > We first outline the previously discussed biases in the work of Gundersen et al. Then, we find that the working hours of the small dataset are relatively constant (14 out of 22 have a value of 40) and not indicative of qualitative documentation in two failure cases. Thus, we compare our method to that of Gundersen instead by correlating successfull outcomes (Experiments reproduced, result label). We find moderate correlations between Implementation and Configuration on all three metrics of success. We also note that these are independent correlations; as stated before we assume there to be a relation that higher cost in multiple dimensions to have a joint correlation is reproducibility success. We however did not find a sufficiently robust statistical model to measure this on our dataset. We also requested the problem ID data from the authors per study, to analyse where our methodology did not sufficiently cover documentation, to highlight concrete limitations of our methodology.
> > > >
> > > > As this is clearly a highly important subject, we first intended to include this in the main body of the paper. Space limitations however would force us to cut substantial parts of this section, and thus reducing its value (More than we were willing to accept). Thus, we have placed in the appendix and have referenced it in Section 5. We are very interested in your opinion on this decision.
> > > >
> > > > # LLM-based Analysis
> > > >
> > > > Our apologies for not stating this clearly; the verification of LLM outputs would be similar to the effort of this work, we intended to highlight that this would be one of the efforts required aside from creating a robust and reliable LLM pipeline.

---

> > > > > ### Comment · Reviewer_wy9e · 2025-11-27
> > > > >
> > > > > Dear authors,
> > > > >
> > > > > ## Correlation Between Reviewers
> > > > > Thank you for the addition of the sections on the selection of second reviewers. The section seems to cover this topic exhaustively. As you have said, there is no perfect solution either way. While there is certainly some degree of bias, it is now well-documented, and it seems like appropriate measures to mitigate the effect have been taken.
> > > > >
> > > > > ## Comparison to Gundersen
> > > > > The addition of table 4 and the whole section A.3 is much appreciated, and it is nice to see that there is a moderate negative correlation between the cost and successful reproductions. The decision to not test against working hours directly is well motivated. While the effect is not particularly strong, the trend is consistent with the expectation, the missing correlation to the data dimension is explainable through the choice of studies, and the difference in distributions is large enough that one can expect several other effects to influence the result. I especially appreciate that the authors made the effort of reaching out to Gundersen et al. and compare the different problem types with the numeric categories of this paper.
> > > > >
> > > > > A minor nit: The section has an incomplete or malformed sentence at line 983, and table 4 marks one of the p values in bold.
> > > > >
> > > > > ## Rating
> > > > > The addition of a comparison to a study which reproduces experiments significantly strengthens the paper. While it is unfortunate that this does not fit into the main paper, it is understandable that authors cannot reorganise large parts of their manuscript around these results. The effort that was spent on showing the correlation is much appreciated, and the diligence with which the data analysis has been conducted is commendable. I thank the authors for the discussion and will raise my score.

---

> > > > > > ### Author Response · Authors · 2025-11-28
> > > > > >
> > > > > > We would like to thank the reviewer for their time, the thorough reading of our manuscript, and the interactive feedback that allowed to substantially improve our manuscript. We have corrected the mentioned mistakes you mentioned in our new version of the manuscript.

---

### Official Review · Reviewer_fGe7 · 2025-10-30

**Soundness:** 2
**Presentation:** 2
**Contribution:** 2
**Rating:** 4
**Confidence:** 4

**Summary:**

This paper explores seven top machine learning and AI conferences/journals to assess the cost of reproducing approximately 50 papers per year between the years of 2022-2024, resulting in an analysis of 918 applicable reviews out of 1061 sampled papers. To conduct this cost-based analysis, the authors initially suggest five dimensions of Implementation, Data, Configuration, Experimental and Expertise; however, they omit Expertise due to their identification of it as a 'noisy' dimension during the second review. Overall, their results highlight that ML conferences typically have the lowest cost of reproducibility compared to AI conferences and Journals. The topic of reproducibility is timely and well chosen due to its importance in verifying scientific contribution and the paper provides a few recommendations alongside its cost framework, for improving reproducibility standards across ML, AI and journal publications.

**Strengths:**

1. The paper identifies cost as an overlooked and understudied dimension of reproducibility studies, offering a novel framework for modelling the five dimensions of cost of published papers.
2. The survey ambitiously reviews 918 applicable papers across seven of the top venues in the AI/ML domain and provides insights into their cost, which provides a large-scale overview of reproducibility cost.
3. Practical insights are offered into how to reduce reproducibility costs, which include using visible reproducibility checklists, appendices and the introductions of reproducibility reviewers at conferences. I believe that these could be impactful in helping to curb reproducibility issues in publications; however, I wonder if they go far enough with regard to the importance of the issue as highlighted in the paper.

**Weaknesses:**

1. **Reducing overstatements**: 918 applicable papers are analysed in the body of the paper, and this should be represented over the number of 1061, which is highlighted in the abstract.
2. **Fixed cost scale of 1-10**: Placing an upper bound on the cost feels as though it could be misrepresentative of how frequently we see extremely high cost papers. Especially in dimensions such as implementation, where a score of 10 is quite frequent. Having looked at the cost scale in Appendix A Table 3, I see no clear rationale for this threshold. Could you provide further reasoning for this?
3. **Balanced representation across venues**: How do you arrive at the statement L461) '150 per venue is statistically robust' across all venues, as each venue has a large variance in the number of publications. For example, NeurIPS has far more papers per year than JMLR, as you recognise in (L996) ' JMLR 2024 only had 49 papers', so it does not seem fair that each venue was not represented proportionally, which could greatly impact the findings.
4. **Second Review**: The use of first authors to review their own work initially strikes me as an odd decision due to potential bias towards providing a lower cost for their own paper. I think the second review is a very important element of this study; however, it does not seem clear why 5.01% is a sufficient amount of second reviews. I worry that trends may change if more second reviews were conducted. Also, the second reviews were not uniformly sampled from all venues, so some venues could be better/worse represented than others in the second review stage, which can greatly impact findings.
5. **Omission of Expertise dimension from Table 3**: It is stated that this dimension was removed from analysis due to disagreements between first and second reviews which appears fair, however, it is important to see how the rubric for this dimension differs from others to observe if the dimension was removed as a result of subjectivity of the particular dimension's evaluations or a property of the sample collected for the first and second review.
6.  **Background Section**: The background feels overly verbose, while it is important to focus on the distinction of definitions for reproducibility, it feels that the content of the section does not add to the overall content of the paper and that this could be made more concise with a fuller discussion in the appendix. Furthermore, it feels more apt to bring forward discussions in the appendix, such as that by Poldrack (2019), which directly discusses reproducibility cost.
7. **Minor points** Table 1 caption appears incorrect, it states that AAAI/ICML have more than 150 publications reviewed, but AAAI only has 142. It might be helpful to highlight that the rubric is an indirect measure of cost earlier in the paper. Line 897 typo of 'rpresnt' which should be represent.

**Questions:**

1. The statement (L996) ' JMLR 2024 only had 49 papers' seems odd to me, as Volume 25 (January 2024 - December 2024)  has far more papers than the 49 discussed in the paper, please can you provide how you obtained this number?
2. How would a larger proportion of second reviews would lead to different results on the agreement between ratings?
3. Can you add a statement on whether you accept or reject your main hypothesis in the main paper?

---

> ### Author Response · Authors · 2025-11-17
> **Rebuttal [1/3]**
>
> We would like to thank the reviewer for the thorough analysis of our manuscript, and detailed feedback.
>
> 1. _Reducing overstatements: 918 applicable papers are analysed in the body of the paper, and this should be represented over the number of 1061, which is highlighted in the abstract._
>
> This is a fair observation, we have adapted the number in the abstract accordingly and marked the change in red.
>
> 2. _Fixed cost scale of 1-10: Placing an upper bound on the cost feels as though it could be misrepresentative of how frequently we see extremely high cost papers. Especially in dimensions such as implementation, where a score of 10 is quite frequent. Having looked at the cost scale in Appendix A Table 3, I see no clear rationale for this threshold. Could you provide further reasoning for this?_
>
> We designed a scale of 1-10 for ease of interpretability for the reader. Although an upperbound is set at ten, the guidelines are generally created such that different studies can be evaluated on the same scale of quality; the guidelines shown in table 1 (formerly table 3 of the appendix) show various reasons cost can be increased, however in each dimension they can rarely exceed the cost of ten for a study; not each ruleset is applicable to each study. Thus we found overall in our study that we did not have to apply a maximum to the given cost for a study, as it followed (in practice) directly from the guidelines that exceeding the limit is rarely possible.
>
> With regards to the implementation dimension specifically, as addressed in your question, the guidelines for example enable us to increase the cost in 9 cases; however, as code is either present or not, that normally only four or five guidelines are applicable to a study. For example, studies that introduce a new algorithm makes (detailed) pseudo code much more applicable; those introducing a new model architecture or framework make design and overviews applicable. The absence of both is rarely applicable, and the only case where this cost could exceed ten (assuming that no other implementation documentation is present). So although it is theoretically possible in this dimension, we believe that there is no directly added value of representing this separately; a cost of ten is to represent a nearly complete absence of documentation and find this both to be the case for those scoring ten and those sligthly exceeding this.
>
> 3. _Balanced representation across venues: How do you arrive at the statement (L461) '150 per venue is statistically robust' across all venues, as each venue has a large variance in the number of publications. For example, NeurIPS has far more papers per year than JMLR, as you recognise in (L996) ' JMLR 2024 only had 49 papers', so it does not seem fair that each venue was not represented proportionally, which could greatly impact the findings._
>
> We chose our (simple) sampling strategy based on the Central Limit Theorem; small subsamples can be representative of a larger population if the subsample is large enough (150) if sampled uniformly at random. Hence, we find each _subset_ of each _venue_ representative, and conduct the majority of the analysis based on splitting these dimensions (hence the name of 5.2. "Analyses per **venue** and dimension'). However, we still find that this is a very valid point when it comes to making assumptions about the _entire_ population of published studies; our dataset is not _proportionally_ representative in this regard as noted by the reviewer. This currently occurs in two measurements in section 5.2; the public data rate and the public implementation rate (and subsequent darkpattern estimate). To mitigate this over/under representation, we determined the corrected weighted percentages based on the number of publications per venue per year. We have added a section for clarification on this subject in Appendix A, and a note on the subject at the end of section 6 (discussion). The corrected percentages of section 5.2 and section 6 are marked in red.

---

> > ### Author Response · Authors · 2025-11-17
> > **Rebuttal [2/3]**
> >
> > 4. _Second Review: The use of first authors to review their own work initially strikes me as an odd decision due to potential bias towards providing a lower cost for their own paper. I think the second review is a very important element of this study; however, it does not seem clear why 5.01% is a sufficient amount of second reviews. I worry that trends may change if more second reviews were conducted. Also, the second reviews were not uniformly sampled from all venues, so some venues could be better/worse represented than others in the second review stage, which can greatly impact findings._
> >
> > This is partially a misunderstanding; the first review was acquired by the first author of _this_ work. We have corrected this sentence to clarify in red. The second reviews the volunteers could select from were curated to ensure spread across venues and years; we ensured that each venue received at least five secondary reviews, and that each year of each venue at least had one secondary review. We have adapted the text in 5.1. in red to reflect this.
> >
> > With regards to the sufficiency of the sample size; although acquiring a larger sample size is always a good thing, we aimed for this rate to ensure our calculations would be a substantial proportion of our entire dataset, and would allow us to make statistically significant findings to the objectiveness and robustness of our methodology. Meta-level studies are very demanding in terms of resources, and is a recurring issue in the field. To place our sample size into context; Werner et al. (2024) [1] used a single study, Gundersen & Kjensmo (2017) [2] sampled 400 papers, Gundersen et al. (2025) [3] used 22, Raff (2019) [4] had a sample size of 255 for their analysis [4,5].
> >
> > We do not believe that increasing the sample size will have a substantial impact on the outcomes of 5.1. Based on this clarification, and that provided in the previous question, answer your question with regards to our statistical approach to the methodology? We are most interested in your opinion on this, such that we can adapt our manuscript accordingly.
> >
> > 5. _Omission of Expertise dimension from Table 3: It is stated that this dimension was removed from analysis due to disagreements between first and second reviews which appears fair, however, it is important to see how the rubric for this dimension differs from others to observe if the dimension was removed as a result of subjectivity of the particular dimension's evaluations or a property of the sample collected for the first and second review._
> >
> > The largest issue with the expertise dimension was the lack of a clear ruleset, and rather than providing a clear point scheme the second reviewer could use to determine the scale, this was rather a loose set of instructions of what elements to take into account, summarised as follows;
> > - How many diﬀerent (sub)ﬁelds within AI are being touched upon?
> > - How well is the problem introduced, or are we expected to already understand the problem?
> > - How much mathematics / logic / proofs are presented, and how well is each introduced?
> >
> > This was mainly to illustrate to the reviewer that we should factor into account how much knowledge the reader is already expected to have, versus being able to grasp the work on its own without relying on external works. It became quickly apparant to us that the lack of these rules were not supportive for any sound findings with our methodology, and thus removed it from the study. This is described in section 5.2. We did not include it in the table, as tangible guidelines were not present in this dimension. We did not wish to spend too much text on the subject either w.r.t. the page limit, hence it is only briefly noted on. We can however include a discussion on this in the appendix. Would that, in your opinion, strengthen our manuscript?
> >
> > 6. _Background Section: The background feels overly verbose, while it is important to focus on the distinction of definitions for reproducibility, it feels that the content of the section does not add to the overall content of the paper and that this could be made more concise with a fuller discussion in the appendix. Furthermore, it feels more apt to bring forward discussions in the appendix, such as that by Poldrack (2019), which directly discusses reproducibility cost._
> >
> > We value this feedback highly, and have adapted our manuscript accordingly. We have reduced the discussion on Goodman et al. (2016) and Gundersen (2021) substantially in the main body, and moved the extended discussion to the appendix.

---

> ### Author Response · Authors · 2025-11-17
> **Rebuttal [3/3]**
>
> 7. _Minor points Table 1 caption appears incorrect, it states that AAAI/ICML have more than 150 publications reviewed, but AAAI only has 142. It might be helpful to highlight that the rubric is an indirect measure of cost earlier in the paper. Line 897 typo of 'rpresnt' which should be represent._
> The count the caption refers to is the total count (i.e. 142+9 for AAAI for example); we have adapted the caption to reflect and clarify this. Please let us know if you find this answer sufficient! With regards to the typo on line 897; We have corrected this and marked the change in red.
>
> # Questions
>
> 1. _The statement (L996) ' JMLR 2024 only had 49 papers' seems odd to me, as Volume 25 (January 2024 - December 2024) has far more papers than the 49 discussed in the paper, please can you provide how you obtained this number?_
> At the time of sampling, only 49 published papers were found for Volume 25 on JMLR.org. This is not clearly specified in our manuscript, and we have adapted this statement accordingly in red. Thank you for noticing this contrariety, and our apologies for this error.
>
> 2. _How would a larger proportion of second reviews would lead to different results on the agreement between ratings?_
> We do not believe that increasing the sample size would substantially impact the findings; we ensured a substantial part of our dataset is independently verified and have taken spread across years and venues into account. Most importantly, we have also made sure that our sample size is large enough for statistically robust findings to the objectiveness of our metric. However, due to the nature of the methodology of our work, validation of our methodology is indispensable to the interpretation of the outcomes of our work; reviewer wy9e suggested to correlate our methodology with the outcomes of Gundersen et al. (2025). We are interested to hear your opinion on such an analysis, and do you believe this could supplement our secondary reviews to determine the objectiveness of our methodology?
>
> 3. _Can you add a statement on whether you accept or reject your main hypothesis in the main paper?_
> We have added a statement at the end of section 6, marked in red, to summarise that we can only partially accept our hypothesis.
>
> # References
> [1] Werner, Luisa, et al. "Reproduce, replicate, reevaluate. the long but safe way to extend machine learning methods." Proceedings of the AAAI Conference on Artificial Intelligence. Vol. 38. No. 14. 2024.
>
> [2] Gundersen, Odd Erik, and Sigbjørn Kjensmo. "State of the art: Reproducibility in artificial intelligence." Proceedings of the AAAI conference on artificial intelligence. Vol. 32. No. 1. 2018.
>
> [3] Gundersen, Odd Erik, et al. "The unreasonable effectiveness of open science in ai: A replication study." Proceedings of the AAAI Conference on Artificial Intelligence. Vol. 39. No. 25. 2025.
>
> [4] Raff, Edward. "A step toward quantifying independently reproducible machine learning research." Advances in Neural Information Processing Systems 32 (2019).
>
> [5] Raff, E. (2021). Research Reproducibility as a Survival Analysis. Proceedings of the AAAI Conference on Artificial Intelligence, 35(1), 469-478. https://doi.org/10.1609/aaai.v35i1.16124

---

> ### Comment · Reviewer_fGe7 · 2025-11-24
> **Reviewer Response to Author Rebuttal (1/2)**
>
> I appreciate the authors taking the time to respond to the concerns raised by my review and the concerns of other reviewers. With regard to issues in formatting and clairty, I think the paper has been improved. However, I still have issues with the core methodology of the paper, which have also been highlighted by other reviewers, and I would appreciate further discussion on this.
>
> **1. Overstatements:**
>
> Thank you for modifying the statements in the work to accurately reflect the number of papers reviewed.
>
> **2. Upper-bound on dimension costs:**
>
> Thank you for further explaining the rationale surrounding dimension costs and the low probability of a cost exceeding 10. However, you do admit that “a cost of ten is to represent a nearly complete absence of documentation and find this both to be the case for those scoring ten and those sligthly exceeding this.” showing that there are some cases that the score of 10 is exceed which contradicts the point you make that “in our study that we did not have to apply a maximum”. Can you please clarify if there were instances that exceeded the cost of 10? If there were, why are they not represented in the data or, at least, discussed in the appendix?
>
> **3. Proportional Results Representation:**
>
> I appreciate the addition of the Appendix Section A.1 ASSUMPTIONS ON THE ENTIRE POPULATION, as well as the edits to the caption of Table 2.
>
> I have a question regarding a statement in Appendix Section A.1: “The estimated public implementation rate, measured in our dataset, 62.13%, corrected to 70.52%% after weighing the sources per year. This is the most substantial change” Why do the authors feel this occurs?
>
> **4. Second review:**
>
> Thank you for clarifying that all first reviews were conducted by the first author of this paper. I am concerned, however, that a more diverse set of reviewers was not used for the initial reviews. Although I recognise that increasing the number of reviewers can be costly, the use of only one first reviewer (n=1) raises the possibility of bias. I do not feel that any responses in the rebuttal have resolved this core issue.
>
> The issue of limited secondary reviewers is significant and has been noted by most reviewers. After reading the responses to comparable concerns, including on this review, I remain unconvinced. In reply to reviewer CjNg, you stated that the methodology was robust because "Our research network is quite diverse." This is anecdotal; a proper evidentiary basis would require a survey of secondary reviewers demonstrating a broad spectrum of expertise, which is missing from the paper. Furthermore, I concur with wy9e that relying on second reviewers from the authors' own network introduces bias due to correlated perspectives.
>
> Furthermore, I feel there is a contradiction in your responses - for example, in this response you state that the second reviews are statistically significant, but then to reviewer CjNg you state “each venue was at least receiving five secondary reviews and at least one per venue per year.”. Here, you acknowledge that for each year there is at least one secondary review per venue per year -  this furthers my concern, while the authors may argue 5.01% is generally representative of all reviews, having some years at some venues with only a sample size of 1 secondary review and claiming statistical significance is not sound. Given that the first reviews were conducted entirely by the first author of this paper, the only costly element of this would be the second review process, and I do not feel as though enough resources have been allocated to ensure that this is a robust analysis.
>
> It is very important that the findings of a reproducibility study are in themselves reproducible and free from bias.

---

> ### Comment · Reviewer_fGe7 · 2025-11-24
> **Reviewer Response to Author Rebuttal (2/2)**
>
> **5. Omission of the Expertise dimension from Table 3:**
>
> Here, I think a better second reviewer selection process could have been prioritised to establish if the issue is with the subjectivity of questions or disparity in second reviewer quality. For example, a selection criterion for second reviewers that provides insights into how many papers second reviewers have published, the number of first-authored papers published by the second reviewer, and their rating of their own mathematical aptitude. If there is a minimum requirement for second reviewers based on such questions, it can ensure that reviewers have the required expertise to assess all dimensions.
>
> Furthermore, I find it difficult to think that only this dimension has subjectivity. For example reviewer CjNg highlighted the dark pattern emergence that can follow from the documentation dimension such as “partially missing code or a large amount of well formatted content that obvisgates important parameters for reproducibility.” - the authors response regarding evaluation of if the “code is "easy to follow”” or "to re-implement the method from scratch?" also feels highly subjective and dependant individual reviewer skill. As such, I think that a larger sample size of the second review may actually uncover less agreement for first and second reviews, especially if reviewers are sampled randomly (under minimum expertise requirements) outside of the author's own network. I request that a discussion on this be provided in the appendix.
>
>
> **6. Background:**
>
> I appreciate the authors bringing forward the work that explicitly discusses the cost of reproducibility provided by Poldrack (2019). The background has improved and is now more centred on the topic of the work.

---

> ### Author Response · Authors · 2025-11-26
> **Response to reviewer fGe7**
>
> Thank you for taking the time for your extensive response and feedback on our manuscript.
>
> # Upper-bound on dimension costs
>
> This is a mistake on our behalf. We meant to clarify that it is **theoretically** possible in one case of our methodology for a cost to exceed ten. We however did not find any instance where this is the case, and thus never had to apply a maximum. We have clarified this in the appendix. Does this answer your question? Our apologies for the unclarity.
>
> # Proportional Results Representation
> When balancing the rates, we re-weighted the impact of each venue to the population based on their proportion to the overall population. As JAIR and JMLR have substantially fewer publications yet a much lower implementation rate than for example the ML conferences (ICLR, ICML, NeurIPS), their effect on the aggregated representation was substantially reduced and thus shifted the estimate upwards. Thank you for this question, we have updated the text in A.1. to include this information.
>
> # Second review
> Based on your feedback on this subject (and other reviewers as you noted) we have added a section to the appendix (A.2.) describing the variety of our research network. We have listed the recorded primary research directions of the volunteers, to demonstrate their broad spectrum of expertise.
>
> Based on your answer regarding the sample size and the note on our response to reviewer CjNg, we would like to clarify; we aimed to ensure each source used is representatively subsampled of our original sampling strategy. The reason that we find this is representative is not based on a year-by-year basis. We analyse the secondary review from a perspective of our methodology, and only use this dataset for this overall purpose. Thus we evaluate the independent reproducibility of our own methodology, and for this we deem a sample size of 46 reasonable based on the central limit theorem. We completely agree with the reviewer that making claims with these second reviews on single years of venues would absolutely not be statistically sound, and apologise for this confusion.
>
> Based on the feedback of wy9e, we have also added another section to our manuscript (Appendix A.3.) to determine the relation of our method with regards to reproduction attempts, to further shed light on this subject. We applied our method to the papers that were reproduced in Gundersen et al. (2025) and determined the correlation between our methodology and ours, and found moderate correlations between our dimensions and the success metrics defined in Gundersen. We also requested the problems encountered by the original authors per reproduction attempt and use this to highlight concrete limitations of our methodology. We are interested in the reviewers opinion, as it came up in a discussion on the same subject.
>
> # Omission of the Expertise dimension from Table 3:
>
> We agree that it is highly important that our secondary reviewer are unbiased and representative, however we disagree with these selection criterions. We selected our secondary reviewers based on their active involvement in research (doctoral students, post-docs and professors), and thus a common reader (and applier) of such works, and allowed reviewers to pair with studies that align with their interests to ensure alignment with the content of the paper and their own area of expertise.
>
> With regards to expertise: we did not mean to imply that expertise was the only dimension that had subjectivity, rather it was the one most substantially affected by it.
>
> On the note by CjNg, the message there was aimed towards whether we evaluated the actual code delivered by the authors. We emphasise here that "to re-implement the method from scratch?" is the general spirit of what we are evaluating the quality of documentation for (What do we aim to use the documentation for), and is not a concrete question asked for the reviewer to answer during the evaluation. We evaluated each study on the contents of the code by verifiying the described methods were actually present in the code, and whether these were readable based on the contents of the readme and the presence of comments in the code. This we do agree with the reviewer has room for subjectiveness, but is a quite small impact on the overall scoring. Nonetheless, we found it important to seperate well documented code from ill-documented code and that this should be represented in the evaluation.
>
> Based on your feedback we have included a new section in the appendix (A.1.). Thank you for your valuable feedback on this subject, and have aimed to highlight these important subjects in our manuscript; we have included in this section a discussion on the secondary review regarding the sample size and these potential biases, and have marked these changes in red.

---

### Official Review · Reviewer_wVap · 2025-10-31

**Soundness:** 3
**Presentation:** 3
**Contribution:** 3
**Rating:** 6
**Confidence:** 5

**Summary:**

The paper presents a comprehensive study on the reproducibility factors of papers published in top ML conferences. The paper introduces a notion of "cost" - different from monetary cost, this cost tracks the amount of roadblocks one could face reproducing a paper by solely relying on the paper and associated released artefacts. The authors construct an annotation guideline for cost on various metrics, and annotate a large amount of papers published in the conferences. The authors present several interesting statistics on the state of reproducibility for multiple conferences.

**Strengths:**

- Experimental methodology is well designed and executed
- Sound analysis and results on the reproducibility issue in multiple conferences and journals
- This work highlights multiple aspects of the reproducibility issues - what I found most interesting is the high % of broken links, and how ICLR, ICML and NeurIPS (flagship ML conferences) have lower implementation cost than other venues and journals. As the authors point out, this is likely a byproduct of strong reproducibility checklists and guidelines adopted by these conferences.
- The fact that award-winning papers also have similar costs as other papers is an interesting outcome.

**Weaknesses:**

- My biggest gripe about the empirical data presented in this paper is the analysis is primarily done by a single reviewer (the first author). Only 5% of the papers were reviewed by different annotators (although the paper does a good job identifying the challenges in such evaluation). While the volume of papers reviewed is much higher than Raff et al or Gundersen et al, the high volume is also a significant source of bias by being annotated by a single reviewer. However, the complexity of annotating this kind of work is very high, so getting more annotations might also be prohibitively costly.
- The definition of cost doesn't consider software deprecation, which is an important aspect to keep in mind while reproducing prior work and is often ill-documented in the repositories.
- The cost estimates perhaps could have been done according to the subcategory of research. Certain popular domains (RL) can likely have lower cost compared to papers introducing frontier models.
- These days the difference in quality of papers in different conferences is slowly becoming less, as its increasingly the same set of people submitting (or resubmitting) the papers. The more interesting metric of tracking cost could have been on time - do we see an increasing/decreasing or mostly constant rate of cost for papers published over the years? This could have helped us understand the state of reproducibility better.

**Questions:**

- Does data cost change with time? I'd assume older papers have lower data cost given the datasets they used to train on are readily available now
- L359-360: the data cost is less costly, so how this is "in conflict" with higher public data rate? shouldn't it be "in agreement"?
- In the same vein, how is JMLR data cost higher even if the public data rate is high? I cannot understand how these two metrics wont be correlated with each other
- In L423, "4.22% of all empirical studies the implementation link is either empty or broken", however in L375 authors note "6.59% of all implementation urls are either empty.. " why are these two values different?

*Suggestions*

- Regarding the conclusion (L482), there is a dedicated venue for publishing reproducibility issues and improvements in ML, the Machine Learning Reproducibility Challenge (MLRC), which could be added here.
- Add links to the ML Reproducibility Checklists
- Re L441: another reason could be why ICLR has similar cost as NeurIPS despite not having stricter reproducibility requirements is the historic use of open reviewing and rebuttals. Curious what the authors think about this.

*Typos*

- L445: ~"do no allow"~ -> "do not allow"

---

> ### Author Response · Authors · 2025-11-17
> **Rebuttal [1/2]**
>
> We would like to thank the reviewer for taking the time to read our work and for providing us with clear and constructive feedback to improve our manuscript.
>
> # Overall feedback
>
> ## Presented Empirical Data
>
> The high requirement of resources for representative outcomes is a struggle throughout the meta research field of reproducibility. To illustrate this struggle in the field and place our work into context with a few other sample sizes: Werner et al. (2024) [1] only used a single paper for their analysis. Gundersen & Kjensmo (2017) [2] used a sample size of 400, Gundersen et al. (2025) [3] a sample size of 22. Raff (2019) [4] sampled 255 papers between 1984 and 2017 for and was used for two seperate studies [4,5]. We have aimed to ensure statistical representative results in our work, but resource demands are highly straining for meta studies. Reviewer wy9e suggested to determine the correlation between the outcomes of Gundersen et al. (2025) [3] and our methodology. There are several caveats with regards to biases; the outcomes have a different timeframe, does not include attempts on papers without public data and has selected the most highly cited papers which is less representative of the entire population of published studies. This has thus far moved us against including this in our work. Would such an analysis, in your opinion, strengthen the understanding of robustness of our method and manuscript in this regard (with the stated biases in mind)? This is a relatively small experiment and could easily be included in our work.
>
> ## Software deprecation
> This subject has received less attention in our work, as we focus on how easy the implementation documentation is to understand and reproduce, rather than how easy it is to use the supplied implementation. However, it is still partially present in our guidelines; we consider 'installation documentation', where the authors specify which software dependencies are being used, to determine how complete the implementation is (See table 1, formerly table 3, first row). This indirectly considers software deprecation, e.g. determining which versions are being used and thus which are deprecated is possible through this, but not to the full extent you pointed out. However, it is important to consider that we conducted our study from the viewpoint of the authors; what are they responsible for and have control over s.t. they can improve the documentation? It would be a harsh requirement to hold them responsible for software deprecation if these dependencies were not developed by them, but plays a role in reproducibility nonetheless especially with regards to older studies.
>
> ## Cost estimates per Subcategory
> The cost estimates were not done per subcategories of research (as there is an abundance of subfields within AI), but during development of the methodology we did consider how each varies and more importantly what they have in common, to create a generalisable guideline. This is for example strongly reflected in our data dimension guidelines; synthetic data generators (such as environments in RL) have completely independent guidelines on how they should be evaluated and what important factors are for qualitative documentation. Although we do not wish to claim that we have covered each subfield entirely (once again, abundance), we found overall that our method generalises well to the population, and measured this as well through our secondary reviewing process.
>
> ## Trends over time
>
> Although initially considered an angle of analysis for this work, the main factor of why such an analysis was not conducted was due to scale; we acquired ~150 reviews per venue, which resulted in a sizable effort to do. This number per venue is a robust sample size, but an analysis per year is substantially less (about 50 per venue per year). Furthermore, when initially exploring such possibilities, we also found that the time range of this study was too narrow; we aimed to evaluate how documentation quality with regards to reproducibility has changed over the recent years, but by doing so we only found weak trends that would only produce weak or uninsightful conclusions. But we completely agree with the reviewer that this is a highly interesting view; however due to the significant effort that such an analysis would take, we believe it warrants an independent study. Does this answer satisfy your question on why we did not conduct an analysis over time? We are very interested to hear your opinion on this matter and would like to dedicate a section in Appendix A to this subject.

---

> > ### Author Response · Authors · 2025-11-17
> > **Rebuttal - Questions & Suggestions [2/2]**
> >
> > # Questions
> >
> > - _Does data cost change with time? I'd assume older papers have lower data cost given the datasets they used to train on are readily available now._
> > As discussed previously, we decided against doing a year-on-year analysis due to sample sizes and time frame. However, we find an extremely high public data rate in our study indicating that studies are often using publicly available datasets in recent research. With regards to older papers, we look to Gundersen et al. (2025); they analysed 30 empirical studies from 2012, 2014 and 2016 (Bias: based on citation count), but only included those with publicly available data; they found 22 out of 30 studies to use public data ($73.33\%$). This would indicate that the usage of public data has increased over time, rather than older works publishing their formerly private dataset. However, the bias we noted on the work of Gundersen et al. (2025) should not be taken lightly before drawing hard conclusions on this matter, and we believe our reasoning should be used as motivation for further investigation.
> >
> > - _L359-360: the data cost is less costly, so how this is "in conflict" with higher public data rate? shouldn't it be "in agreement"?_
> > Although data cost is overall quite low, we wanted to highlight that such a high public data rate can lead to an expectation of an even lower data cost. We however found that this correlation is perhaps weaker than may be expected. We have slightly adapted our phrasing to clarify this; we find these two correlated with one another, but simply expected a stronger correlation and want to highlight this to the reader, as well as illustrate to the reader why this may be the case.
> >
> > - _In the same vein, how is JMLR data cost higher even if the public data rate is high? I cannot understand how these two metrics wont be correlated with each other._
> > Continuing on our answer of the previous question, and the adaptation described; we expected a higher public data rate to be very tightly correlated to lower data cost overall, yet found conflicting evidence that the venue with the highest rate is also the least well documented in this dimension. We found this an important aspect to highlight to the reader; nonetheless, data cost is found overall to be low. Does this clarification answer your question? We have not adapted this paragraph (Data paragraph of section 5.2.) thus far, and are interested to hear your opinion on this s.t. we can clarify this better in our manuscript.
> >
> > - _In L423, "4.22% of all empirical studies the implementation link is either empty or broken", however in L375 authors note "6.59% of all implementation urls are either empty.. " why are these two values different?_
> > We agree with the reviewer that these two values are confusing, but not conflicting; The first proportion is calculated over all empirical studies, e.g. how many studies contain a broken URL, the second is over the studies that actually _contain_ an URL. We agree with the reviewer that these two statements are confusing, and have adapted the first statement to use the same proportion. The change is reflected in our manuscript, and highlighted in red. We chose to use the second proportion (i.e. over the URLs), as papers without any URL cannot have a broken URL to begin with. We would be interested to hear your opinion on our choice, such that we can make sure our findings are presented as accurately and fairly as possible. Note that this value has been adjusted slightly; originally we reported this value across our dataset. However, for better interpretability, we have now calculated this metric across our dataset including weights per venue s.t. each venue is proportionally represented based on the number of empirical publications per venue per year. We have clarified this in Appendix A.1.
> >
> > # Suggestions
> >
> > - We have updated the conclusion to include the MLRC; thank you for bringing this venue to our attention. We have marked the changes in red.
> > - These checklist links are present in footnote 2, section 6 as hrefs for each venue. Would you prefer if these were present as full links for clarity? Currently they are present as hrefs to due to page limitations. Are these the checklists you refer to in this suggestion, or are there other checklists that are not yet included into our manuscript that you refer to?
> > - We have adapted this paragraph in section 6 in red. To briefly summarise; ICLR is unique in its usage of public reviewing process. We do not know how this knowledge affects the authors in writing the original submission, nor do we know how often the public interacts and provides feedback during the review process. We believe, based on our findings w.r.t. ICLR, that this warrants an analysis of its own, to determine the relation between this open review process and the quality of documentation for reproducibility.

---

> ### Author Response · Authors · 2025-11-17
> **References**
>
> # References
> [1] Werner, Luisa, et al. "Reproduce, replicate, reevaluate. the long but safe way to extend machine learning methods." Proceedings of the AAAI Conference on Artificial Intelligence. Vol. 38. No. 14. 2024.
>
> [2] Gundersen, Odd Erik, and Sigbjørn Kjensmo. "State of the art: Reproducibility in artificial intelligence." Proceedings of the AAAI conference on artificial intelligence. Vol. 32. No. 1. 2018.
>
> [3] Gundersen, Odd Erik, et al. "The unreasonable effectiveness of open science in ai: A replication study." Proceedings of the AAAI Conference on Artificial Intelligence. Vol. 39. No. 25. 2025.
>
> [4] Raff, Edward. "A step toward quantifying independently reproducible machine learning research." Advances in Neural Information Processing Systems 32 (2019).
>
> [5] Raff, E. (2021). Research Reproducibility as a Survival Analysis. Proceedings of the AAAI Conference on Artificial Intelligence, 35(1), 469-478. https://doi.org/10.1609/aaai.v35i1.16124

---

> ### Author Response · Authors · 2025-11-28
>
> We would like to use this comment as a friendly reminder to the reviewer, as the deadline for us to implement their feedback draws close.

---

### Official Review · Reviewer_CjNg · 2025-11-06

**Soundness:** 1
**Presentation:** 3
**Contribution:** 2
**Rating:** 2
**Confidence:** 4

**Summary:**

This paper attempts to quantify the cost of reproducibility by reviewing the available documentation  of 1061 papers from a number of the most prominent AI and machine learning conferences. Cost is measured across multiple dimensions, namely Implementation, Data, Configuration and Experimental procedure. A list of guidelines are used to score different works along these dimensions, with 46 papers recieving two reviews. Machine learning conferences were found to require lower reproducibility costs than other venues. Best paper awards were not any more or less reproducible than other papers.

**Strengths:**

* This is an extremely important topic within the scientific community and therefore equally important to evaluate within the field of AI.
* This work evaluates a large number of papers and does reveal some interesting insights into reproducibility patterns within the AI and ML conferences evaluated. For example, the paper illustrates the lack of detailed documentation in the absense of public code.
* The paper is also generally well written.

**Weaknesses:**

The current procedure to determine cost is heavily reliant on a number of variables that rely on the subjective estimation of the effort required by researchers to reproduce research and is not measured against a more objective measure (such as hours needed by researchers to reproduce experiments). More specifically, for implementation cost, the research question is posed

*"Given the documentation shared by the authors on a new method, how much effort would it be to re-implement the method from scratch?".*

I'm not sure that this question is answered in this paper since the gap between  *reading*   experimental documentation and *reproducing* it can be significant, as some challenges only become clear upon attempting to run code/implement experimental proceedures. A more convincing motivation for the scoring procedure proposed in this work would be to determine the correlation the proposed cost with the hours taken to reproduce work, perhaps for a random subsample of data. Though this is clearly a more challenging task, I think this is quite important as, in the same way that the authors have observed links that are available in a paper but lead to nowhere, there could be similar dark patterns in documentation, partially missing code or a large amount of well formatted content that obvisgates important parameters for reproducibility.

I also find it challenging to justify decoupling the hardware cost from the reproducibility cost as I am inclined to disagree with the assertion that accounting for variation in computation is more closely aligned with outcome reproducibiliy, espectially in an era where large-scale models are presented as superior due to their sheer size. Such assertions cannot be tested in a tractable mannor without access to similarly large-scale computational resources. I also do have some questions on the sampling of papers for review as there does appear to be some level of sampling bias, which I have detailed in my questions below.

**Questions:**

1. p. 5 *"We evaluate each dimension based on guidelines that allow for the expression of subjective variation between studies."* => Can you elaborate on what this means?


2. p. 6. *"Due to the difficulty of the task, the effort required for reviewing and the challenge of finding proficient reviewers, initially, only a single review was acquired for each paper by the first author"*  => Was the first review for each paper conducted by the first author themselves?


3. p. 6 *"To mitigate this limitation, we acquired a second review for 46 (i.e. 5.01%) out of the 918 papers from independent researchers in our network of PhD candidates and post-doctoral researchers. We let our 14 volunteers select up to five papersfor secondary review based on their expertise and personal interest"* => Could there be a selection bias here in terms of both the composition of reviews (within a single network) and the papers which are selected by them? Isn't it likely that members of a similar network want to read papers based on similar topics?

---

> ### Author Response · Authors · 2025-11-17
> **Rebuttal [1/2]**
>
> We would like to thank the reviewer for taking their time to read our manuscript and providing us insights that allow us to improve our manuscript.
>
> # Objective measure
>
> Although there is some subjectivity to our metric, we designed our method to follow guidelines that are applicable quite universally, such as the presence of (pseudo) code, availability of data or parameter values; there are certain parts of the method more vulnerable to subjectivity. To mitigate this, or at least understand to what degree this affects the outcome of the experiment, we conducted a secondary review for a substantial part of the dataset, and found strong ICC for the majority of the dimensions, where the mean absolute distance is substantially less than one score point. Based on this outcome, we adapted our analysis as well, and only drew conclusions where the statistically significant differences also showed substantial shifts in terms of the test value (i.e. measured distance between the distributions). We would like to ask the reviewer, in the light of the usage of secondary reviewers, motivates them to find the measurements still too subjective?
>
> We also noted your mention of the objective measurement of hours needed for reproduction. Reviewer wv9e also suggested that conducting an analysis on the studies used for Gundersen et al. (2025), where hours necessary for reproduction was also measured. Would such a supplementary analysis, determining the correlation between our method and the results of working hours in Gundersen et al. strengthen our manuscript and provide better insights to what degree our measures are objective? We currently have not done this thus far, based on the fact that there are several biases in this dataset that could reduce the effectiveness of this analysis:
>
> 1. These papers are less representative of the entire population of published research (Only the most highly cited were used)
> 2. Only studies with public datasets were considerd (e.g. ~6% of the population is not represented in their dataset).
> 3. The papers were selected outside of our time range (2012/2014/2016 vs 2022-2024 in our work).
>
> As this is a relatively small analysis, we would like to hear your opinion on this so that we can adapt our manuscript accordingly.
>
> ## Implementation cost
>
> Although there is a substantial difference between reading documentation and reproducing the actual work, the main aim of this work was to estimate the _quality_ of the documentation. It is obvious that this plays a central role when reproducing a method; when the implementation (code) is not clearly documented (or not present at all) it has a major impact on the amount of labour required by the independent investigators. Some reproducers find that true reproduction only occurs when the _code_ is implemented independently from scratch, rather than re-using the code of the authors. Thus we framed the question to the strongest extent ("to re-implement the method from scratch?").
>
> Although some challenges of reproduction only become apparant upon attempting this, we would like to emphasise to the reviewer that _the code of each study was inspecting when present_ (See guidelines table 1, formerly table 3 of the appendix), to determine if the majority of the method was covered and to what degree the code is "easy to follow" based on comments and other documentation regarding the code (readmes etc.). Hence, our work reflects to what extent the authors have _enabled_ independent reproduction w.r.t. the implementation and reduced the workload of independent investigators. We feel that there is some confusion here, and have adapted section 4.2. slightly in red to reflect that the code (when provided) was investigated for readability and completeness. We would like to ask the reviewer for their opinion on this change, as we would like to clarify this confusion throughout our manuscript.

---

> > ### Author Response · Authors · 2025-11-17
> > **Rebuttal [2/2]**
> >
> > # Decoupling the Hardware cost
> >
> > We would like to note that we believe this subject into two separate categories; the availability of compute, and the accounting for variation in outcomes due to varying hardware.
> >
> > ## Availability of Compute
> > As described in the third paragraph of the introduction, we decided against including this as we desired to understand the role of documentation rather than the availability of compute. We would like to clarify this. The main motivation to exclude this is the fact that the authors do not have any responsibility to make compute available to independent investigators, nor do we believe authors should be 'punished' for having access to compute resources that may be unavailable to the majority of the community; this would hold back advancements in science unnecessarily.
> >
> > ## Variation in Hardware
> > We believe we have phrased this too concisely in our manuscript (Appendix C, paragraph 1). We have adapted this paragraph as follows; the effects of hardware on experiment outcomes are substantial, but we believe this can be accepted as long as the results _uphold the same interpretations_ and thus the same hypotheses. This similarly holds true for non-deterministic methods; if authors do not document the seeds used for example, we find this not to be a substantial issue, as long as the outcomes under different circumstances uphold the same hypothesis.
> >
> > We would like to ask the reviewer if, based on the third paragraph of the introduction and Appendix C, this clarifies our view on the relation between compute/hardware and reproducibility
> >
> > # Questions
> >
> > _p. 5 "We evaluate each dimension based on guidelines that allow for the expression of subjective variation between studies." => Can you elaborate on what this means?_
> >
> > We mean to express briefly here that the guidelines are designed to capture varying types of studies from various subfields; different guidelines are applied for example to studies that use environments or synthetic data generators versus static datasets for example. We have rephrased this sentence as follows (and adapted the manuscripts accordingly in red); "We evaluate each dimension based on our guidelines that cover a wide range of topics allowing for the expression of a variety of topics across studies and subfields."
> >
> > Our apologies for this confusion; does our answer satisfy your question?
> >
> > _p. 6. "Due to the difficulty of the task, the effort required for reviewing and the challenge of finding proficient reviewers, initially, only a single review was acquired for each paper by the first author" => Was the first review for each paper conducted by the first author themselves?_
> >
> > The review was conducted by the first author of _this_ work; we have clarified this in section 5.1. in red.
> >
> > _p. 6 "To mitigate this limitation, we acquired a second review for 46 (i.e. 5.01%) out of the 918 papers from independent researchers in our network of PhD candidates and post-doctoral researchers. We let our 14 volunteers select up to five papers for secondary review based on their expertise and personal interest" => Could there be a selection bias here in terms of both the composition of reviews (within a single network) and the papers which are selected by them? Isn't it likely that members of a similar network want to read papers based on similar topics?_
> >
> > The reviews conducted in the original experiment were sampled uniformly at random. Afterwards, we sampled this subset uniformly at random, but limited the choices in this subset s.t. each venue was at least receiving five secondary reviews and at least one per venue per year. Although this is a valid point, we also restricted papers to only receive two reviews; two volunteers were not allowed to select the same paper to review. Furthermore, our research network is quite diverse in terms of subtopics, which enabled us to conduct this experiment with variation and spread. We have adapted section 5.1. slightly to reflect this in red. Does this answer your question? We would love to hear your feedback, and make sure all information discussed here is reflected into our manuscript.

---

> ### Author Response · Authors · 2025-11-28
>
> We would like to use this comment as a friendly reminder to the reviewer, as the deadline for us to implement their feedback draws close.

---

### Official Review · Reviewer_i52Z · 2025-11-10

**Soundness:** 1
**Presentation:** 2
**Contribution:** 1
**Rating:** 2
**Confidence:** 3

**Summary:**

This paper conducts a study of the cost of reproducibility of 918 (empirical) papers from AAAI, IJCAI, ICLR, ICML, NeurIPS, JAIR, and JMLR from the years 2022-2024. The cost of replicability was assessed on a scale from 1-10 and along 5 axes: expertise cost, experimental cost, configuration cost, data cost, and implementation cost. Implementation cost is claimed to be the most expensive for all venues. Configuration cost was the next most expensive overall, followed by data cost, followed by experimental procedure. Expertise appears to not have been reported due to issues with grounding/objectivity of the evaluation. ML conferences were found to have lower costs of replicability compared to more general AI venues.

**Strengths:**

This study represents a significant effort in assessing reproducibility costs of publications in major ML venues. The sample size is large and appears representative. The cost categories are largely reasonable, though I’m a bit uncertain that configuration cost represents a true reproducibility cost distinct from implementation cost, since parameter setting ought to follow from the implementation.

**Weaknesses:**

The paper would benefit from being restructured. A large chunk of the introduction is spent on historical anecdotes, apparently at the expense of  including the scoring methodology in the body of the paper. Since this scoring table significantly affects the interpretation of results, I would expect it to be included in the main body.

I am also not sure what the take away from this paper ought to be. The costs of reproducibility across the 5 dimensions do not share any particular unit of measurement, and have varying maximum values, but they’re compared to one another as though their scores are comparable metrics. While scores within a category and total scores across venues could certainly be compared, I don’t think the observation that implementation reproducibility is most costly can be supported by this methodology.

Comment on presentation -
“We agree with Goodman et al. (2016) on the intuition behind what they refer to a search for the ‘truth’.” but then go on to say, “We also disagree with the reasoning that the search for ‘truth’ is the main motive for reproducibility. We find our answer in Popper (1934) instead: we argue it is the credibility of a method.”
I’m generally not sure what the take away from this paragraph is meant to be, and this point in particular is unclear.

**Questions:**

Please see comments in weaknesses.

---

> ### Author Response · Authors · 2025-11-17
> **Rebuttal [1/3]**
>
> We would like to thank the reviewer for taking the time to provide us with valuable feedback on our manuscript.
>
> # Configuration versus Implementation
>
> Although we agree with the reviewer that parts of the documentation with regards to the algorithm configuration may overlap with any submitted code (repositories), in our definition these two elements do not overlap and the dimensions are designed to be 'orthogonal' to one another. We treat submitted code (Either through repository links or supplementary materials) as _extended documentation_. The documentation of the implementation or configuration is not limited to be only placed in the paper or the code; for example, design choices for the implementation can be found throughout the paper (diagrams, pseudo code or library choices) as well as in the code. The same holds true for the algorithm configuration; this can be detailed in the appendices for example with hyperparameter tables, or be included in the repository with configuration files. The most important difference between implementation and configuration is that these serve two different purposes, and can create two completely disjoint reasons of why a work is easily reproducible or not. Attempting to implement a method versus determining the configuration used to produce the results of the paper are two seperate parts of the process. This does not invalidate the statement of the reviewer however that the parameter setting ought to follow from the implementation; it can be documented there, and often is but can also follow from the main paper or appendices.
>
> # Restructuring the paper
>
> In our manuscript, we dedicated a large section of the introduction (as well as the background section) to explain our view on the definition of reproducibility and the effect that (lack of) documentation has in terms of labour required when independently reproducing a study. The reason that we spent two paragraphs in the introduction on the historical example of Boyle, Hobbes and Huyges is to emphasise to the reader that this subject is neither new nor unimportant; documentation plays a key role to determine whether a study can be considered independently reproducible, yet has received relatively little attention in previous studies that assess this aspect specifically. Hence, we deemed in necessary that this subject is to be studied individually.
>
> We agree with the reviewer that, although motivating a study clearly to the reader is highly important, it should not be at the expense of the methodology. This choice was originally made due to page limitations. We have adapted our manuscript accordingly based on this feedback as well as reviewer wy9e, and marked the changes in red. We have also adapted the reference to the table slightly, to allow the reader to better interpret the results based on our methodology.

---

> > ### Author Response · Authors · 2025-11-17
> > **Rebuttal [2/3]**
> >
> > # Take away of the paper
> >
> > We would like to thank the reviewer for this concrete feedback. We made changes to the abstract, background, and conclusions to clarify and improve the phrasing of message. The main takeaway of our work is summarised as follows;
> >
> > - Reproducibility is an essential part of the scientific method, and has been discussed extensively over the years due to the ongoing reproducibility crisis. However, previous studies that assess the state of reproducibility in published work have given relatively little attention to the importance of documentation, yet its role is indispensable. We summarise the phenomenon of increased labour due to lacking documentation as the 'cost of reproducibility'.
> > - As most major venues recently have undertaken steps to stimulate authors to provide better documentation (e.g. reproducibility checklists), we would like to determine how this has affected the research communities from these respective venues. We hypothesise that venues with stronger requirements, will have overall better documented publications and are thus less costly to reproduce. To answer that question empirically, we design a scoring methodlogy to quantify the quality of documentation across 5 dimensions, to determine which key elements lack qualitative documentation in published empirical research. We find that:
> >
> > 1. There is a significant difference in reproducibility cost between AI venues and ML venues, which is also reflected in a dark pattern w.r.t. implementation links, provide insights into what may cause this, and how this can be (partially) mitigated.
> > 2. Reproducibility checklist play a smaller role to documentation quality than we had hypothesised, as illustrated by the differences between NeurIPS and ICLR for example. This was an unexpected finding, where we provide insights into what may cause this, and how venues can either increase the effectiveness of checklist (stronger regulation through for example an independent reviewer, enforce that code is submitted through supplementary material for archival purposes), or through more positive motivation as illustrated in our next point:
> > 3. There is no significant difference between award winning papers and 'average' papers, indicating that reproducibility plays no significant role when adjuciating awards. This means that the community gets little stimulation to write documentation that leads to low cost reproducibility. Furthermore, award winning papers serve as exemplary studies; by making reproducibility a key part of this process, they can inspire the community to follow suit.
> >
> > The main takeaway should of course directly be clear to any reader and our abstract was written to serve exactly that purpose. We would like to ask to the reviewer whether they could specify and clarify to us which sections of our work cause this confusion, as we find soundness and presentation extremely important for our work, and would like to improve this wherever possible.
> >
> > ## Unit of Measurement
> >
> > The unit of measurement in our work is a representation of the quality of documentation with regards to its expected impact to the labour required for reproducing a study, where we have created a scale from 1 to 10 for ease of interpretation. This is defined in the paper, Section in 4.2. To the best of our knowledge, this is clearly defined here, and would like to ask the reviewer what in our work suggests the varying maximum values, such that we can improve the clarity and presentation of the manuscript. With regards to the comparison; the values found per dimension are only compared per dimension, to avoid unfair comparison between these seperated scales. Could you clarify what you mean with "compared to one another as though their scores are comparable metrics"? We have sought to treat the data from our study with care, such that the outcomes can be used for constructive conclusions, and would like to improve the presentation to avoid such misleading implications in our manuscript.
> >
> > ## Observation on Implementation reproducibility
> >
> > Although we do state in our work that we find reproducibility the most costly dimension, we believe that this partially overstated; we find in our results that implementation and configuration are the most ill-documented (e.g. highest scores) when compared to the other dimensions. This does not mean that, using our methodology, we can derive a direct impact for the amount of labour (e.g. working hours) needed, and refer to the findings of Gundersen et al. (2025) [7] to determine that implementation has a bigger impact on reproducibility cost as a dimension when compared to configuration. Thank you for pointing this out; to clarify this in our manuscript, we have rephrased our statements in Section 5.2. and Section 6 to describe the distributions rather than insinuate an ordered relation between the dimensions. We have marked these changes in red.

---

> ### Author Response · Authors · 2025-11-17
> **Rebuttal [3/3]**
>
> # Presentation
>
> The original intent of the first statement, was to point out that we find the motivation in the work of Goodman et al. [1] quite convicing, but their conclusion lacking a clear link with the philosophy of the scientific method. Hence, we looked to Popper [2], to fill this gap. We understand however, that these statements, which are so closely related to one another, are confusing and have adapted our manuscript to improve the presentation of the paper; we have removed the first statement, to make the paragraph more concise.
>
> The overall objective of this paragraph is to give the reader insights into our literature study surrounding the subject of reproducibility. As it is an ill defined subject [3,4,5,6], as discussed in the first paragraph in section 2, we deemed in necessary to provide reasoning to the why we settled on this definition specifically. The main take away of this paragraph, is as follows:
>
> We find various definitions of reproducibility, and provide reasoning as to why even the closest definitions are in some regards lacking. The main take away is that we settle on an adapted or derived version of that in Gundersen (2021) [6], where we explain why we do not use any categories/degrees of reproducibility in this work, and will be used for our methodological setup in Section 4. In section 4 we concisely reformulated the definition from Gundersen [6]. The reason that we spent a substantial part of the paper on this subject, is that we cannot rely on a single source to define 'reproducibility', and find it essential for understanding our methodological setup. We believe that this lack of clarity in our work is (partially) caused by the physicial separation of section 2 and 4; thus to ensure to the reader that the link between this paragraph, and our methodology, we have adapted the paragraph in red. Furthermore, based on the feedback of reviewer fGe7, we have moved a large part of this literature discussion to the appendix.
>
> We would like to ask the reviewer if this clarification of this paragraph, and the subsequent adaptation of this part of our manuscript has brought the desired improvement of the presentation and contribution of this paragraph to our work.
>
> # References
>
> [1] Steven N. Goodman, Daniele Fanelli, and John P.A. Ioannidis. What does research reproducibility mean? Science Translational Medicine, 8(341):341ps12–341ps12, June 2016.
>
> [2] Karl Popper. The logic of scientific discovery. Julius Springer, Hutchinson & Co, 1934.
>
> [3] Ramal Moonesinghe, Muin J. Khoury, and A Cecile Janssens. Most published research findings are false-but a little replication goes a long way. PLoS Medicine, 4:218–221, 2007.
>
> [4] Daniel J. Simons. The value of direct replication. Perspectives on Psychological Science, 9 (1):76–80, 2014.
>
> [5] Edward Raff, Michel Benaroch, Sagar Samtani, and Andrew L. Farris. What do machine learning researchers mean by “reproducible”? In Proceedings of the 39th AAAI Conference on Artificial Intelligence, volume 39, pp. 28671–28683, 2025.
>
> [6] Odd Erik Gundersen. The fundamental principles of reproducibility. Philosophical Transactions of the Royal Society, 379(2197):20200210, 2021.
>
> [7] Odd Erik Gundersen, Odd Cappelen, Martin Mølnå, and Nicklas Grimstad Nilsen. The unreasonable effectiveness of open science in ai: A replication study. Proceedings of the 39th AAAI Conference on Artificial Intelligence, 39(25):26211–26219, April 2025. doi: 10.1609/aaai.v39i25.34818.

---

> ### Author Response · Authors · 2025-11-28
>
> We would like to use this comment as a friendly reminder to the reviewer, as the deadline for us to implement their feedback draws close.

---

### Meta-Review · Area_Chair_jfSu · 2025-12-03

**Summary:**

This paper estimates the cost of reproducing research in ML papers at various venues, making several findings.  The paper finds that award winning papers are not less costly to reproduce, and ML venues have lower reproduction cost than AI venues or journals.  Reviewers raised the following criticisms: (1) poor paper structure, (2) lack of interesting and actionable takeaways, (3) subjectivity of the proposed metric, (4) possibility that the proposed metric does not truly capture or well-approximate the amount of effort it would take to reproduce work, (5) lack of accounting for computational costs of reproduction, (6) annotations by a single reviewer, (7) no accounting for software deprecation, (8) lack of categorical breakdown, (9) upper bound on cost where there shouldn’t be one

**Reviewer Concerns:**

The authors made minor updates to address (1).  I am not bothered by (5) or (9).  However, (8) may actually be a very large problem because different venues receive different types of submissions and hence if there are big differences across subfields, the results may be uninteresting.  Without further examination of (8) and also a more robust annotation system, I am unconvinced by the author’s responses.

**Reviewer Scores:**

Reviewer scores started off as 2, 2, 4 (presumably would not raise score), 6 (presumably raised to 8).  Given the serious problems identified by reviewers and not sufficiently addressed by authors, I assume the reviewers who gave the paper scores of 2 and 4 would not have raised their scores.  While I think this paper is a cool idea, and I am strongly supportive of the development of this work, I agree with these reviewers that the paper is not ready for publication.

---

### Decision · Program_Chairs · 2026-01-26

Reject